# C-type lectin receptor CLEC4A2 promotes tissue adaptation of macrophages and protects against atherosclerosis

Inhye Park [1], Michael E. Goddard[1,5], Jennifer E. Cole [1,5], Natacha Zanin[1], Leo-Pekka Lyytikäinen [2], Terho Lehtimäki [2], Evangelos Andreakos [3], Marc Feldmann[1], Irina Udalova [1], Ignat Drozdov[4] & Claudia Monaco [1✉]

Macrophages are integral to the pathogenesis of atherosclerosis, but the contribution of distinct macrophage subsets to disease remains poorly defined. Using single cell technologies and conditional ablation via a *LysM*[Cre+] *Clec4a2*[flox/DTR] mouse strain, we demonstrate that the expression of the C-type lectin receptor CLEC4A2 is a distinguishing feature of vascular resident macrophages endowed with athero-protective properties. Through genetic deletion and competitive bone marrow chimera experiments, we identify CLEC4A2 as an intrinsic regulator of macrophage tissue adaptation by promoting a bias in monocyte-to-macrophage in situ differentiation towards colony stimulating factor 1 (CSF1) in vascular health and disease. During atherogenesis, CLEC4A2 deficiency results in loss of resident vascular macrophages and their homeostatic properties causing dysfunctional cholesterol metabolism and enhanced toll-like receptor triggering, exacerbating disease. Our study demonstrates that CLEC4A2 licenses monocytes to join the vascular resident macrophage pool, and that CLEC4A2-mediated macrophage homeostasis is critical to combat cardiovascular disease.

[1] Kennedy Institute of Rheumatology, Nuffield Department of Orthopaedics, Rheumatology and Musculoskeletal Sciences, University of Oxford, Oxford, UK. [2] Department of Clinical Chemistry, Fimlab Laboratories and Finnish Cardiovascular Research Center-Tampere, Faculty of Medicine and Health Technology, Tampere University, Tampere, Finland. [3] Biomedical Research Foundation, Academy of Athens, Center for Clinical, Experimental Surgery and Translational Research, Athens, Greece. [4] Bering Limited, London TW2 6EA, UK. [5] These authors contributed equally: Michael E. Goddard, Jennifer E. Cole. ✉email: claudia.monaco@kennedy.ox.ac.uk

Macrophages support organ function and immune responses to tissue damage and disease[1]. Ontogeny and organ-specific tissue signals drive the phenotype and function of resident macrophages through activation of specific transcription factors[2,3]. In some organs, monocytes replenish the resident macrophage pool, adapting their transcriptional signatures to the tissue microenvironment[4,5]. The arterial wall is structurally divided into three layers: the intima, the media, and the adventitia. In the steady state, two distinct macrophage populations reside in the intima[6] and adventitial layers[7]. Both require colony-stimulating factor 1 (CSF1) for their survival and originate from embryonic precursors[6,8]. The majority of lymphatic-vessel endothelial hyaluronan receptor-1 (Lyve1)+ adventitial macrophages are replenished by bone marrow (BM)-derived monocytes after birth[8], while arterial intimal resident macrophages are replaced by monocytes during early atherosclerosis[6]. The former regulate collagen production in smooth-muscle cells and prevent unfavourable arterial remodelling[7] while the latter are responsible for cholesterol uptake and initiation of vascular inflammation by producing interleukin-1β (IL-1β)[6].

Atherosclerosis is the most prevalent chronic inflammatory disease in man and the main driver of cardiovascular disease[9]. During atherogenesis, monocytes are recruited into the subendothelium of the artery, where they give rise to lesional macrophages[10]. Single cell technologies have revealed three broad macrophage states in murine atherosclerosis, comprising resident, inflammatory, and lipid-driven subsets[11–14]. Genetic lineage tracing studies showed that monocytes have the capacity to differentiate into most macrophage subpopulations during atherosclerosis[15]. The fate of monocyte-derived macrophages is influenced by cues from surrounding tissue, as indicated by the conversion of monocytes into resident-like macrophages during disease regression[16]. The mechanisms of macrophage tissue adaptation in vascular health and disease are currently unknown.

CLEC4A, also known as dendritic cell immunoreceptor (DCIR), is one of the few C-type lectin receptors (CLR) that contain an immunoreceptor tyrosine-based inhibition motif (ITIM)[17]. The ITIM domain recruits Src homology region 2 domain-containing protein tyrosine phosphatase 1 (SHP1) and SHP2, that downmodulate tyrosine phosphorylation signalling[18]. Clec4a polymorphisms are positively associated with human autoimmune diseases[19,20]. CLEC4A2, the murine homologue of CLEC4A, protects from experimental arthritis and autoimmune encephalomyelitis by negatively regulating activation of dendritic cells (DC)[21,22]. Mice lacking Clec4a2 are resistant to excessive inflammation and tissue damage during hepatitis, cerebral malaria, and tuberculosis[23–25]. Collectively, this evidence implies an immunomodulatory role for CLEC4A2, however, its functions in macrophage biology and atherosclerosis are unknown.

In this paper, we reveal that the expression of CLEC4A2 identifies a subset of vascular resident macrophages with atheroprotective functions. CLEC4A2 promotes appropriate tissue adaptation of vascular macrophages ensuring homeostasis of the vessel wall and thus combating atherosclerosis.

## Results

**The expression of *Clec4a2* is associated with a homeostatic signature in vascular resident macrophages.** We previously identified two broad subsets of macrophages in the atherosclerotic aorta, comprising the CD206+CD169+ and CD11c+CD44+ subsets by mass cytometry[12]. Here, we compared the composition of vascular-macrophage subsets in the steady state and atherosclerosis using mass cytometry (Supplementary Fig. 1a). In the aorta of C57BL/6 J wild type (WT) mice, CD206+CD169+

macrophages represented $96.3 \pm 0.4\%$ of total macrophages (Supplementary Fig. 1b). Atherogenesis promoted a reduction in the representation of this aortic macrophage population to $73.7 \pm 2.0\%$ in high fat diet (HFD)-fed apolipoprotein E-deficient (ApoE)$^{-/-}$ mice. Conversely, CD11c+CD44+ macrophages significantly increased with atherosclerosis from $3.8 \pm 0.4\%$ in WT to $26.3 \pm 2.1\%$ in ApoE$^{-/-}$ (HFD) mice.

The spatial distribution of these macrophage populations was assessed by microscopy and mass cytometry. CD206+ macrophages were mainly detected in the adventitia in healthy and atherosclerotic aortas (Supplementary Fig. 1c,d), whereas macrophages co-expressing CD11c and CD64 were predominantly located in the intima of the atherosclerotic aorta (Supplementary Fig. 1d). To clarify the macrophage heterogeneity across arterial compartments, we performed mass cytometry on single cells from separated intima-media and adventitial layers of aortas of ApoE$^{-/-}$ mice (Supplementary Fig. 1e). Unbiased tSNE clustering demonstrated that adventitial macrophage clusters expressed CD206, CD169, and Lyve1, while macrophage clusters in the intima-media compartment expressed CD11c and CD44 (Supplementary Fig. 1f,g,h,i).

To further dissect the heterogeneity of vascular macrophages and identify their regulators in atherosclerotic mice, we applied single cell RNA sequencing (scRNA-seq) to CD45+CD11b+ myeloid cells from aortas of HFD-fed ApoE$^{-/-}$ mice, and gene expression was acquired using a Chromium 10X Genomics platform (Fig. 1a; Supplementary Fig. 2a,b). Identification of the myeloid subsets, including macrophages, monocytes (C3: Ly6C+ monocytes, C9: Ly6C- monocytes), DCs (C7), and neutrophils (C4), was based on established marker expression[26,27] (Fig. 1b,c). Eight clusters (C0, C1, C2, C5, C6, C8, C10 and C11) expressed the core macrophage genes *Adgre1, C1qa, Fcgr1*, and *Sepp1* and lacked DC and monocyte genes (Fig. 1d). C2 consisted of inflammatory monocyte-macrophages defined by the expression of *Ccr2, Cd14* and *Tlr2* (Fig. 1d). C8 exhibited the previously identified gene expression signature of TREM2$^{high}$ macrophages (*Trem2, Cd9, Lgals3, Itgax*, and *Mmp12*)[11]. Genes that were highly associated with proliferation (*Stmn1, Birc5*, and *Mki67*) were enriched in C10, consistent with the findings of Lin et al.[15]. We also detected macrophages (C11) expressing interferon (IFN)-inducible genes (*Ifit2, Isg15*, and *Irf7*), as previously identified in murine atherosclerotic aortas[15].

Three clusters (C0, C1, and C5) shared resident macrophage genes including *Lyve1, Sepp1, Mrc1, Pf4, Cd163*, and *Folr2*. C1 was defined by the expression of *Cd209b, Cd209f, Cd209g, and Clec4a2* (Fig. 1c,d; Supplementary Data 1). The Gene Ontology (GO) enrichment analysis of C1 assigned functions, including phagocytosis, iron metabolism and antigen presentation (Fig. 1e). C5 (expressing *Lyve1, Cd163, and Clec4a2* but lacking *Cd209* isoforms) specialised in functions such as complement activation, regulation of vascular cells, and cholesterol metabolism. C0 exhibited the gene signature of resident macrophages (*Lyve1, Mrc1, Pf4, Cd163*, and *Folr2*) but also expressed inflammatory and chemokine genes (*Nlrp3, Nfkb1, Ccl7, Ccl2, Ccl24*, and *Cxcl1*). C0 strongly expressed *Mt1, Mt2* and *Hmox1*, which suggests that their activated state may have been induced by exposure to oxidised phospholipids[28]. The GO analysis suggested that C0 was associated with chemotaxis and inflammatory responses. An additional cluster, C6 lacked expression of *Lyve1* and *Cd163* but expressed other resident macrophage genes (*C1qa, C1qb*, and *Pf4*) alongside antigen-presentation genes (*Cd74, H2-Eba*, and *H2-Aa*). The enriched gene signature of C6 assigned pathways associated with antigen presentation and processing and innate immune responses.

Next, we examined the cluster-specific gene lists to identify a regulator of a homeostatic gene signature within the resident

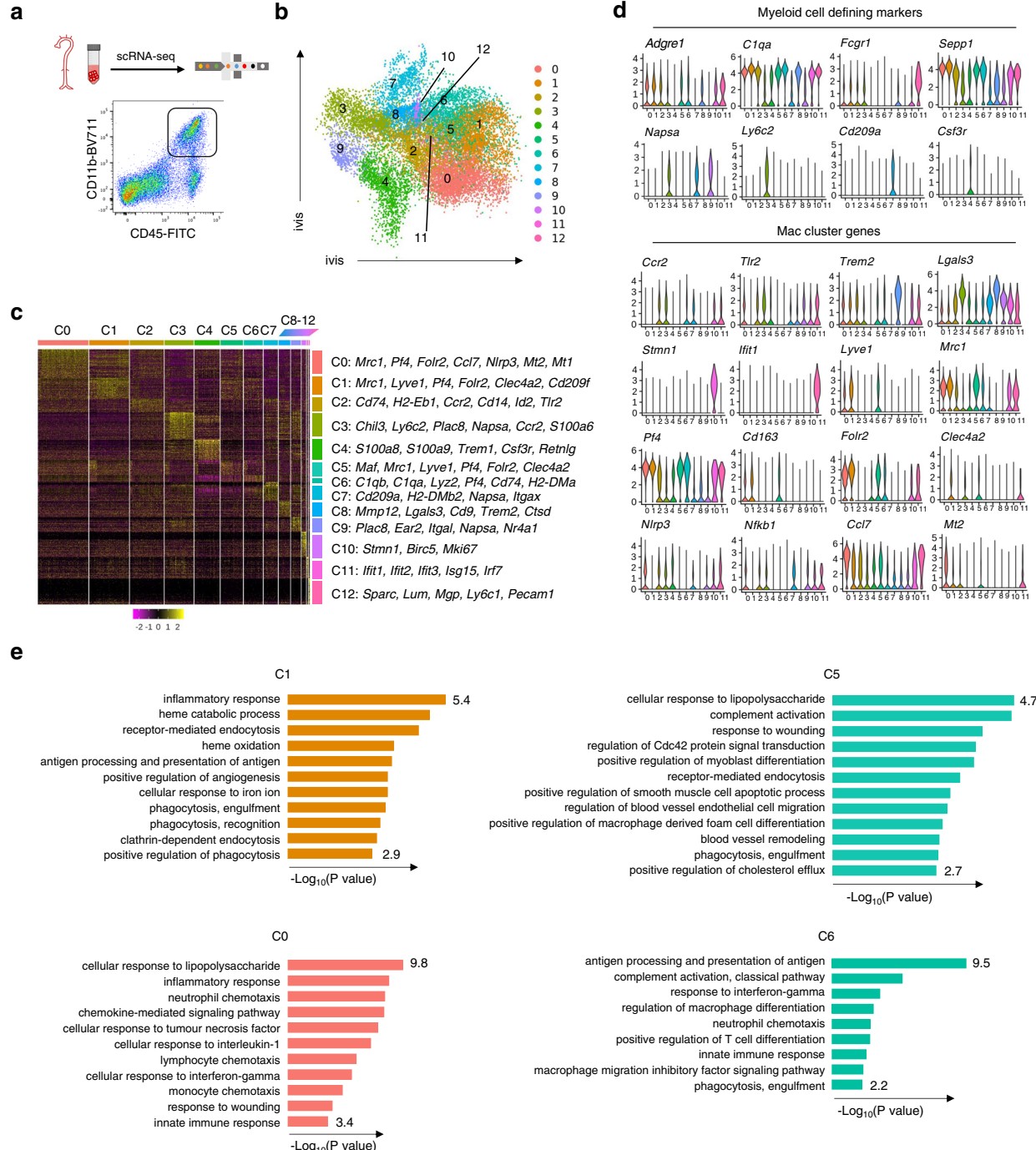

**Fig. 1 Single-cell RNA sequencing reveals that *Clec4a2* expression defines vascular macrophages with homeostatic features in the murine atherosclerotic aorta. a** Gating strategy for CD45+CD11b+ cells from whole aortas of 20–26-week-old *ApoE*−/− female mice fed a high-fat diet (HFD) for 12–16 weeks. **b** ivis clustering of 16,100 vascular myeloid cells analysed by single-cell RNA sequencing (scRNA-seq) using the 10X genomics platform. Cells were pooled from 9 mice in each experiment. Three independent experiments were performed and normalised before concatenation. **c** Heatmap of differentially expressed genes (P < 0.05) in 13 clusters. Cluster 12 expressing stromal genes was excluded from further data analysis. Differentially expressed genes were identified using the two-sided Wilcoxon Rank Sum test with Bonferroni's correction for multiple-hypothesis testing. **d** Violin plots of selected genes in 12 myeloid clusters. **e** The Gene Ontology (GO) enrichment analysis of clusters based on differentially expressed genes in resident-like macrophage subsets (C0, C1, and C5) and C6. Enriched pathways are presented as -Log10 (P values) using topGO analysis. P values were obtained using the one-sided Fisher's exact test without multiple-testing correction. P values < 0.05 were considered significant.

macrophage clusters. Among the differentially expressed genes (P < 0.05), *Clec4a2* was one of the top genes that distinguished two *Lyve1*+*Mrc1*+*Pf4*+ resident macrophage subsets (C1 and C5) from the activated resident subset C0 (Fig. 1c). CLEC4A2 expression in C1 and C5 was negatively associated with the

average expression of 170 macrophage-activation genes, while a weaker correlation was noted in C3 that expressed lower levels of *Clec4a*2 (Supplementary Fig. 2c; Supplementary Data 2). This finding suggests that *Clec4a*2 may have a role in modulating macrophage activation and homeostasis.

**CLEC4A2 is highly expressed by resident macrophages in the aorta, heart, and lung.** The expression of CLEC4A2 and human CLEC4A was reported in neutrophils, monocytes, and DCs in lymphatic organs and blood in WT mice[29] and in human blood[17]. We profiled the cell surface expression of CLEC4A2 in cells from the aorta, heart, and lungs of $ApoE^{-/-}$ and WT mice using mass cytometry. CLEC4A2 was not expressed by non-immune $CD45^-$ cells (Supplementary Fig. 3a). The expression of CLEC4A2 significantly varied according to cell type and tissue of residence. CLEC4A2 was highly expressed by the majority of vascular resident macrophages ($92.4 \pm 0.9\%$ in $ApoE^{-/-}$; $92.0 \pm 1.4\%$ in WT), and intermediate levels of CLEC4A2 were expressed by $62.6 \pm 2.0\%$ of cDC2 in atherosclerotic aortas and by $57.1 \pm 4.8\%$ of cDC2 in WT (Supplementary Fig. 3b,c,d). It was detectable at very low levels in other vascular myeloid cells including $CD11c^+$ macrophages, monocytes, and neutrophils. CLEC4A2 was also expressed by cardiac resident and alveolar macrophages (Supplementary Fig. 3d). In keeping with these findings, purified aortic and alveolar macrophages expressed higher *Clec4a2* gene expression compared with DCs (Supplementary Fig. 3e).

Next, we assessed the regulation of *Clec4a2* expression in macrophages by tissue-derived factors and inflammatory stimuli. Ex vivo conditioned media from murine aortas and hearts significantly enhanced the expression of *Clec4a2* in CSF1-cultured bone marrow-derived macrophages (BMDMs), as well as the expression of *Mrc1* (CD206) (Supplementary Fig. 3f). Conversely, stimulation with lipopolysaccharide (LPS) and IFNγ significantly inhibited *Clec4a2* expression in CSF1-BMDMs, while IL-4 and IL-13 did not alter its expression (Supplementary Fig. 3g).

To understand the relationship between the human homologue of CLEC4A2, CLEC4A, and markers of resident macrophages in human atherosclerotic plaques, we interrogated transcriptomics data from human carotid arterial tissues in the Tampere Vascular Study biobank. We performed Spearman correlation between *Clec4a* and macrophage genes and found that *Clec4a* gene expression was strongly correlated with markers of resident and alternatively activated macrophages (*Cd163, Csf1r, Cd200r1, Clec7a, Tlr7, Scarb1, Cd68, Msr1, Mrc1,* and *Ms4a4a*) (Supplementary Fig. 4). This dataset shows that a close relationship between *Clec4a* expression and macrophages is also conserved within human vascular tissues.

**Diphtheria toxin-mediated cell ablation in $LysM^{Cre+}$ $Clec4a2$-$^{flox/DTR}$ mice targets $CLEC4A2^+$ tissue macrophages.** In order to study the role of $CLEC4A2^+$ macrophages in vascular disease, we generated a mouse strain $Clec4a2^{flox/DTR}$ carrying a construct containing neomycin, diphtheria-toxin receptor (DTR), and eGFP upstream of exon 8 of the *Clec4a2* gene (Supplementary Fig. 5a). This mouse line was further crossed with $LysM^{Cre}$ knock-in mice so that diphtheria toxin (DT)-mediated ablation in cells expressing both genes could be achieved while preserving $CLEC4A2^+$ cDC2, which expressed CLEC4A2 in aortic tissues. The mouse line was viable and displayed no differences in body weight ($Clec4a2^{flox/DTR}$: $28.6 \pm 0.9$ g ($n = 28$) and $LysM^{Cre+}$ $Clec4a2^{flox/DTR}$: $29.4 \pm 0.4$ g ($n = 33$)). The expression of eGFP was detected in neutrophils and to a lesser extent in $Ly6C^+$ monocytes in the BM and blood (Supplementary Fig. 5b), but not in other myeloid cells. In keeping with the data in Supplementary Fig. 3c, gene expression of *Clec4a2, egfp,* and *Hbegf* (DTR) was higher in aortic and alveolar macrophages than DCs and $Ly6C^+$ monocytes from the same tissues (Supplementary Fig. 5c).

Aorta sections immunostained with an antibody against CD64 confirmed macrophage ablation in the adventitia of $LysM^{Cre+}$ $Clec4a2^{flox/DTR}$ mice 24 h after administration of one dose of DT (Fig. 2a). Flow cytometry analysis showed that $CLEC4A2^+$ aortic macrophages were ablated at day 1 (65% reduction), day 2 (80%), and day 3 (75%) post DT injection (Fig. 2b). The aortic macrophage pool was mostly reconstituted at day 5 and fully recovered at day 7, day 14, and day 70 post ablation. An influx of $Ly6C^+$ monocytes was observed post DT administration, peaking at day 3 and waning at day 5, in line with the kinetics of macrophage replenishment (Fig. 2c). No significant DT-mediated depletion was observed in $CD11c^+$ macrophages and neutrophils in the artery at all time points (Fig. 2d,e). CLEC4A2 expression was gradually restored in vascular macrophages from day 7 post ablation (Fig. 2f).

We assessed the effects of DT administration on circulating cells in the BM, blood, and spleen. DT caused a temporary ablation of neutrophils in the BM at 24 and 30 h but the population was restored by 48 h (Supplementary Fig. 5d). In the blood, neutrophils were not significantly reduced by DT treatment nor in the spleen (Supplementary Fig. 5e,f). No depletion was observed in $Ly6C^+$ monocytes in the BM, blood, or spleen (Supplementary Fig. 5g–i).

We studied the feasibility and selectivity of repeated DT treatment by flow and mass cytometry on vascular leucocyte populations from saline or DT-treated $LysM^{Cre+}$ $Clec4a2^{flox/DTR}$ mice. DT-treated mice showed selective vascular macrophage ablation ($92.7 \pm 1.6\%$), without depleting aortic monocytes, cDC1, cDC2, neutrophils, eosinophils, B cells, or T cells (Fig. 2g–l). Similar results were observed in the heart and lung (Supplementary Fig. 5j,k; Supplementary Table 1). No effects of DT treatment were found in circulating cells in the BM, blood, or spleen (Supplementary Table 1).

To trace whether $Ly6C^+$ monocytes repopulate the vascular macrophage pool post depletion, we utilised monocyte-adoptive transfer. Monocytes isolated from the BM of $Cx3cr1^{eGFP+}$ $CD45.1^+$ $CD45.2^+$ mice were transferred into $LysM^{Cre+}$ $Clec4a2^{flox/DTR}$ mice (steady state) and hypercholesterolemic low-density lipoprotein-receptor $(Ldlr)^{-/-}$ mice (CD45.1) after DT treatment (Supplementary Fig. 6a–c). The transferred monocytes were traceable in the blood (Supplementary Fig. 6d,e), and the aortas in the steady state and atherosclerotic conditions (Supplementary Fig. 6f,g). The reconstituted vascular macrophages from both donor and host origins expressed CLEC4A2 (Supplementary Fig. 6f–g). We detected the presence of the donor-origin $GFP^+$ macrophages in the adventitia of $LysM^{Cre+}$ $Clec4a2^{flox/DTR}$ and $Ldlr^{-/-}$ chimeric mice (Supplementary Fig. 6h,i), in keeping with previous parabiosis[8] and fate-mapping[15] studies. In addition, we also confirmed that ablation and replenishment of macrophages occurred in the adventitia of healthy and atherosclerotic mice, leaving lesional macrophages intact (Supplementary Fig. 6j,k).

To investigate the contribution of blood $Ly6C^+$ monocytes and local proliferation to replenishment of vascular macrophages, we performed bromodeoxyuridine (BrdU)-proliferation assays on BM chimeras where the host-derived cells and donor cells were traceable by CD45.1 and CD45.2 expression. $CD45.1^+$ hosts were reconstituted with the BM from $LysM^{Cre+}$ $Clec4a2^{flox/DTR}$ (CD45.2) mice and received DT or saline followed by BrdU injections (Supplementary Fig. 6l). The majority of vascular macrophages in the saline group were donor $(CD45.2^+)$-derived but a small population of host $(CD45.1^+)$ cells (radio-resistant) were detectable, which represented approximately 10% of aortic macrophages (Supplementary Fig. 6m). In the DT group, proliferation rates increased from $9.0 \pm 1.2\%$ to $43.9 \pm 1.9\%$ in the donor origin and $1.2 \pm 0.3\%$ to $8.3 \pm 0.8\%$ in the host macrophages. Collectively, our findings indicate that the replenished resident macrophage pool has a predominant contribution of donor cells.

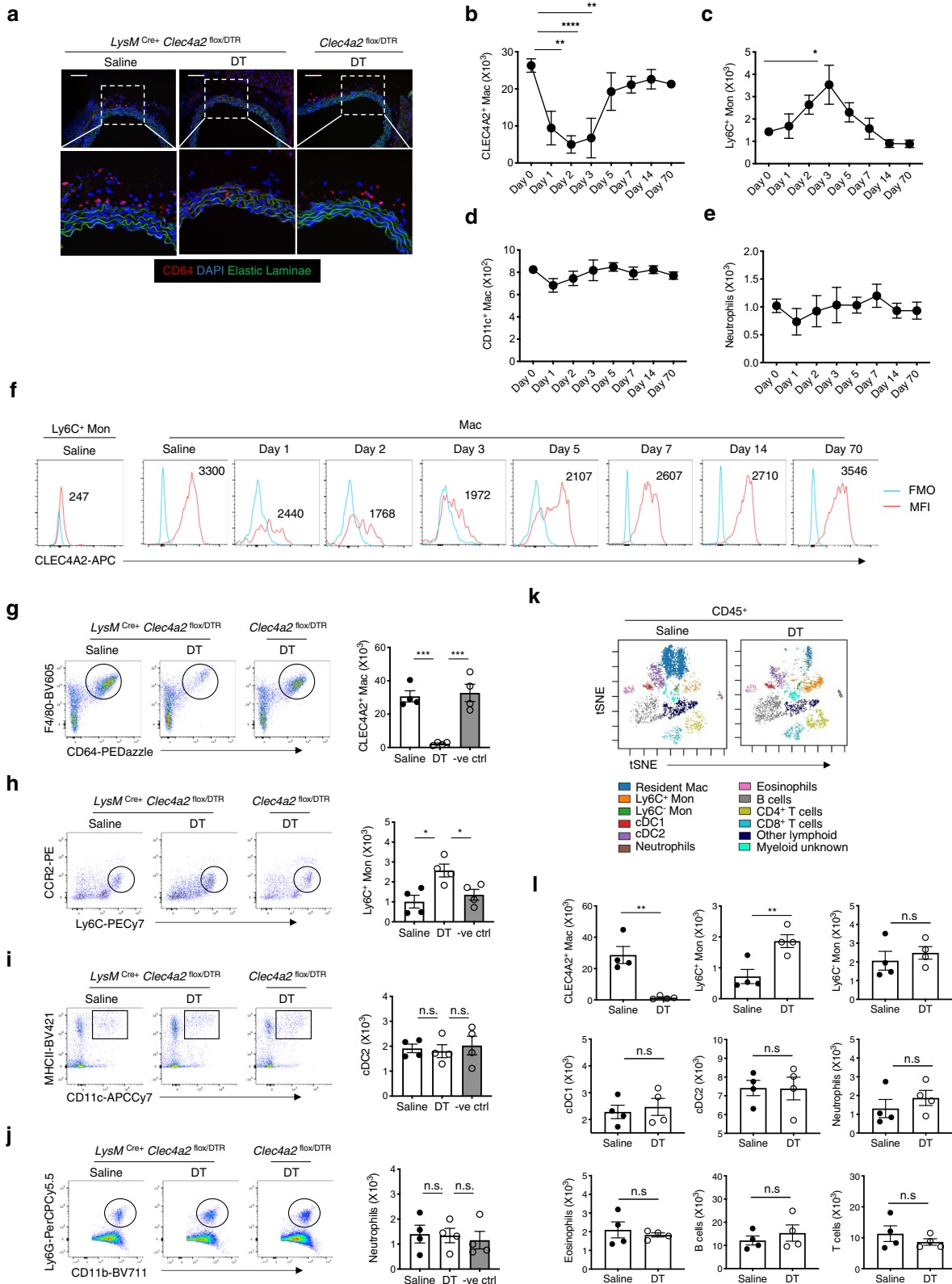

**CLEC4A2⁺ macrophages are endowed with protective functions in atherosclerosis**. Next, we interrogated the function of CLEC4A2⁺ macrophages in atherosclerosis using $LysM^{Cre+}$ $Clec4a2^{flox/DTR}$ mice. $Ldlr^{-/-}$ (CD45.1) host mice received BM transplant from $LysM^{Cre+}$ $Clec4a2^{flox/DTR}$ (CD45.2) or $Clec4a2^{flox/DTR}$ mice (CD45.2). These mice were fed a HFD for 12 weeks to induce

atherosclerosis and received saline or DT treatment fortnightly for 10 weeks (Fig. 3a). Approximately 72.1 ± 6.0% of CLEC4A2⁺ resident macrophages were ablated by DT treatment, whilst aortic CD11c⁺ macrophages, Ly6C⁺ monocytes, Ly6C⁻ monocytes, cDC2, and neutrophils were not affected by DT (Fig. 3b). DT-treated mice developed threefold larger atherosclerotic lesions in

**Fig. 2 _LysM_<sup>Cre+</sup> _Clec4a2_<sup>flox/DTR</sup> mice allow selective ablation of vascular CLEC4A2<sup>+</sup> macrophages. a** Representative images of CD64 (red)-stained thoracic aorta sections from _LysM_<sup>Cre+</sup> _Clec4a2_<sup>flox/DTR</sup> and _Clec4a2_<sup>flox/DTR</sup> mice 24 h after a saline or diphtheria toxin (DT) injection ($n = 4$ mice each; Scale bar: 100 μm). **b–e** Cell numbers of CLEC4A2<sup>+</sup> macrophages, Ly6C<sup>+</sup> monocytes, CD11c<sup>+</sup> macrophages, and neutrophils in whole aortas of _LysM_<sup>Cre+</sup> _Clec4a2_<sup>flox/DTR</sup> mice at 1, 2, 3, 5, 7, 14, and 70 days post administration of DT. Day 0 represents the saline-treated control group (**b–e**: $n = 3–9$ mice in each group, pooled from two to three independent experiments). Ordinary one-way ANOVA with Holm–Sidak's multiple-comparison test. **b** **$P = 0.0011$ (Day 1), ****$P < 0.0001$ (Day 2), **$P = 0.0033$ (Day 3). **c** *$P = 0.0167$. **d** and **e** not significant (n.s.). **f** The mean fluorescent intensity (MFI) of CLEC4A2 expression in vascular macrophages post depletion. FMO stands for fluorescence minus one. **g–l** _LysM_<sup>Cre+</sup> _Clec4a2_<sup>flox/DTR</sup> and _Clec4a2_<sup>flox/DTR</sup> mice received three intraperitoneal injections of saline or DT in a week. Aortic cells were analysed by flow and mass cytometry. **g–j** Representative flow cytometry plots depict macrophages, Ly6C<sup>+</sup> monocytes, cDC2 and neutrophils from aortas of saline or DT-treated _LysM_<sup>Cre+</sup> _Clec4a2_<sup>flox/DTR</sup> mice and DT-treated _Clec4a2_<sup>flox/DTR</sup> mice. Cell numbers of each population in the three groups ($n = 4$ mice; the data are representative of three independent experiments). Ordinary one-way ANOVA with Holm-Sidak's multiple comparisons test. **g** ***$P = 0.0006$ (saline vs. DT and DT vs. negative DT control). **h** *$P = 0.0166$ (saline vs. DT), *$P = 0.0402$ (DT vs. negative DT control). i and j: n.s. **k** Overlaid tSNE plots of vascular leucocytes from saline or DT-treated _LysM_<sup>Cre+</sup> _Clec4a2_<sup>flox/DTR</sup> mice by mass cytometry. Leucocyte populations were identified based on expression of 32 markers. **l** Cell numbers in 9 aortic leucocyte populations across the 2 treatment groups by mass cytometry ($n = 4$ mice; the data are representative of two independent experiments). Two-tailed Student's t-test. CLEC4A2<sup>+</sup> Mac: **$P = 0.0025$, Ly6C<sup>+</sup> Mon: **$P = 0.0011$. All data are presented as mean ± SEM. Source data are provided as a Source Data file.

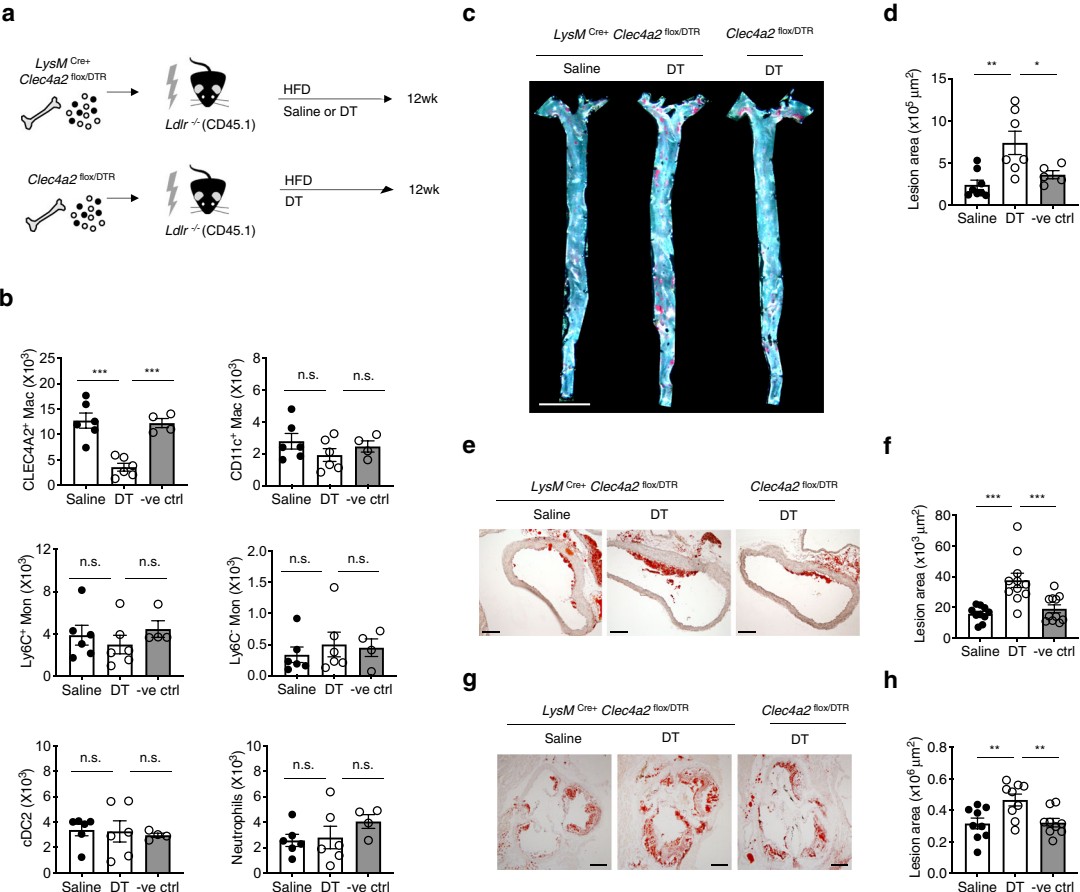

**Fig. 3 CLEC4A2<sup>+</sup> macrophages protect from atherosclerosis. a** Schematic of macrophage ablation in atherosclerotic mice. Bone marrow (BM) cells from _LysM_<sup>Cre+</sup> _Clec4a2_<sup>flox/DTR</sup> and _Clec4a2_<sup>flox/DTR</sup> mice were transplanted into _Ldlr_<sup>−/−</sup> (CD45.1) mice. The chimeric mice received a HFD for 12 weeks and intraperitoneal injections of saline or DT fortnightly for 10 weeks. **b** Cell numbers of resident macrophages, CD11c<sup>+</sup> macrophages, Ly6C<sup>+</sup> monocytes, Ly6C<sup>−</sup> monocytes, cDC2, and neutrophils from aortas for each treatment group (saline and DT: $n = 6$ mice each, negative DT control: $n = 4$ mice; one experiment). Ordinary one-way ANOVA with Holm–Sidak's multiple-comparison test. ***$P = 0.0002$ (saline vs. DT), ***$P = 0.0005$ (DT vs. negative DT control). n.s. not significant. **c** and **d** Representative images and plaque quantification of Sudan IV (red)-stained en face whole aortas of _Ldlr_<sup>−/−</sup> chimeric mice for each treatment group (Scale bar: 5 mm) (saline: $n = 8$ mice; DT: $n = 7$ mice; negative DT control: $n = 5$ mice). Ordinary one-way ANOVA with Holm–Sidak's multiple comparisons test. **$P = 0.0033$, *$P = 0.0348$. **e** and **f** Representative images and plaque quantification of Oil Red O (ORO)-stained sections of ascending aorta of _Ldlr_<sup>−/−</sup> chimeric mice for each treatment group (Scale bar: 200 μm) (saline: $n = 10$ mice; DT and negative DT control: $n = 11$ mice each). Ordinary one-way ANOVA with Holm–Sidak's multiple-comparison test. ***$P = 0.0002$ (saline vs. DT), ***$P = 0.0007$ (DT vs. negative DT control). **g** and **h** Representative images and plaque quantification of ORO-stained aortic root sections of _Ldlr_<sup>−/−</sup> chimeric mice for each treatment group (Scale bar: 200 μm) ($n = 9$ mice each). Ordinary one-way ANOVA with Holm–Sidak's multiple-comparison test. **$P = 0.0080$ (saline vs. DT), **$P = 0.0080$ (DT vs. negative DT control). All data are presented as mean ± SEM. Source data are provided as a Source Data file.

the en face aorta than saline-treated mice (Fig. 3c,d). Similarly, ablation of CLEC4A2⁺ macrophages also exacerbated lesion formation in the ascending aorta and the aortic root (Fig. 3e–h). The levels of serum cholesterol or IL-6 were not altered in all three groups (Supplementary Table 2). Our data demonstrate that the selective ablation of CLEC4A2⁺ vascular macrophages is detrimental to atherogenesis.

**CLEC4A2 exerts a protective function in atherosclerosis by maintaining myeloid homeostasis.** After demonstrating the protective role of CLEC4A2⁺ resident macrophages in vascular disease, we sought to assess how this CLR protects from atherosclerosis. *Clec4a2*$^{−/−}$ mice displayed no discrepancy in numbers and frequencies of cells in lymphatic tissues and blood[21]. We found that WT and *Clec4a2*$^{−/−}$ littermate mice showed no changes in immune cell populations in aortas, lungs and BM in the steady state (Supplementary Table 3). To assess the role of CLEC4A2 in hyperlipidemic conditions, we generated *ApoE*$^{−/−}$ and *ApoE*$^{−/−}$ *Clec4a2*$^{−/−}$ littermate mice and fed them a chow diet or a HFD for 12 weeks to allow for the development of atherosclerosis (Fig. 4a; Supplementary Fig. 7a). No significant differences in body weight or serum cholesterol levels were detected between the two groups (Supplementary Table 4). In the chow-fed mice, we detected no significant differences between the two genotypes in lesion size, arterial size, or necrotic core content in the aortic root (Supplementary Fig. 7a–d). Expression of the mononuclear phagocyte marker CD68 in aortic root lesions was also not altered by CLEC4A2 deficiency in the chow-diet group (Supplementary Fig. 7e). However, we found increased lesion formation in the aortic arch and abdominal aorta of *ApoE*$^{−/−}$ *Clec4a2*$^{−/−}$ mice fed a HFD compared to their *ApoE*$^{−/−}$ counterparts (Fig. 4b). In addition, *Clec4a2* deficiency significantly aggravated lesion development in the aortic root (Fig. 4c). *ApoE*$^{−/−}$ *Clec4a2*$^{−/−}$ mice exhibited an enlarged arterial diameter, indicating outward arterial remodelling (Fig. 4d). Furthermore, *ApoE*$^{−/−}$ *Clec4a2*$^{−/−}$ mice showed a more vulnerable plaque phenotype compared with *ApoE*$^{−/−}$ mice, with a three-fold increase in the necrotic core area (Fig. 4e). *ApoE*$^{−/−}$ *Clec4a2*$^{−/−}$ mice showed increased immuno-positive area for the mononuclear phagocyte markers CD68, CD11c, and CD64 within atherosclerotic lesions (Fig. 4f–h), indicating a higher macrophage content of the lesions.

Next, we investigated whether CLEC4A2 deficiency altered the immune cell composition in the aortas of *ApoE*$^{−/−}$ and *ApoE*$^{−/−}$ *Clec4a2*$^{−/−}$ mice using mass cytometry (Fig. 4i; Supplementary Fig. 7f). Chow-fed *ApoE*$^{−/−}$ *Clec4a2*$^{−/−}$ mice showed a significant reduction in vascular resident macrophages while no changes were detected in other leucocyte populations (Supplementary Fig. 7g,h). When mice developed advanced disease following feeding of a HFD, we consistently found a significant decrease in resident macrophages (Fig. 4j,k). In contrast to the earlier stages of atherogenesis, an increase of CD11c⁺ macrophages, Ly6C⁺ monocytes, and neutrophils was detected in the aorta of HFD-fed *ApoE*$^{−/−}$ *Clec4a2*$^{−/−}$ mice (Fig. 4j,k). No differences in other vascular leucocyte populations were found between the two genotypes. Furthermore, the frequency of immune cells in the blood, BM, spleen and para-aortic lymph nodes was not altered in HFD-fed *ApoE*$^{−/−}$ and *ApoE*$^{−/−}$ *Clec4a2*$^{−/−}$ mice, except for a small reduction in Ly6C⁺ monocytes in the blood of *ApoE*$^{−/−}$ *Clec4a2*$^{−/−}$ mice (Supplementary Table 5). Taken together, CLEC4A2 deficiency results in excessive loss of the vascular resident macrophage pool followed by an increase in lesion size, necrotic core formation, and accumulation of inflammatory myeloid cells.

**CLEC4A2 maintains macrophage identity and proliferation by fostering CSF1-driven differentiation.** Next, we sought to elucidate how CLEC4A2 regulates macrophage biology. We performed poly (A) RNA sequencing (RNA-seq) on CSF1-BMDMs from WT and *Clec4a2*$^{−/−}$ mice. Compared with WT macrophages, *Clec4a2*$^{−/−}$ macrophages displayed a significant downregulation of 241 genes and upregulation of 123 genes (Fig. 5a; Supplementary Data 3). Notably, the set of downregulated genes was associated with macrophage identity and maturation (*Mrc1, Siglec1, Slco2b1, Folr2, Adgre1, Fcgrt, Itgam, Notch1, Cebpa,* and *Irf4*) (Fig. 5b). In contrast, genes regulating immune activation and DCs (*Naip6, Bhlhe40, Id2, Apobec1, Cd80,* and *Cd83*) were augmented in the absence of CLEC4A2. In support of the transcriptomic changes in *Clec4a2*$^{−/−}$ macrophages, 54.6 ± 6.4% of *Clec4a2*$^{−/−}$ CSF1-BMDMs expressed CD206 (*Mrc1*) whereas 74.7 ± 1.5% of the WT counterpart was CD206⁺ (Fig. 5c).

In order to assess the cell-intrinsic role of CLEC4A2 in macrophage differentiation, we cocultured WT and CLEC4A2-deficient cells with CSF1 or CSF2 in a competitive setting (Fig. 5d). In CSF1 cultures, WT cells outcompeted *Clec4a2*$^{−/−}$ cells in their ability to differentiate into CD11b⁺F4/80⁺CD206⁺ macrophages (WT vs. KO: 58.7 ± 4.0% vs. 36.6 ± 3.6%) (Fig. 5e). CLEC4A2 deficiency decreased the proliferation capacity of CSF1-driven macrophages, as lower levels of 5-ethynyl-2′-deoxyuridine (EdU) incorporation were detected in *Clec4a2*$^{−/−}$ cells (Fig. 5f). In contrast, *Clec4a2*$^{−/−}$ BMDMs (CD11b⁺CD11c⁺F4/80⁺MHCII⁻) outcompeted WT cells in the macrophage fraction when cultured with CSF2 (WT vs. KO: 40.5 ± 3.1% vs. 59.4 ± 3.1%) (Fig. 5g) and showed higher proliferation than WT cells (Fig. 5h). Taken together, our data demonstrate that CLEC4A2 fosters monocyte transition to CSF1-dependent macrophages while limiting CSF2-driven differentiation.

**CLEC4A2 promotes tissue adaptation of vascular macrophages in healthy and atherosclerotic aortas.** To determine whether CLEC4A2 could regulate the differentiation of monocytes into resident macrophages in healthy and atherosclerotic aortas in vivo, we performed competitive BM chimera experiments using WT (CD45.1) and *Ldlr*$^{−/−}$ (CD45.1) mice respectively as recipients. These two host mice received 50% BM from WT (CD45.1 CD45.2 heterozygous) and 50% *Clec4a2*$^{−/−}$ (CD45.2) mice (Fig. 6a). The *Ldlr*$^{−/−}$ chimeric mice received a HFD for 12 weeks to induce atherosclerosis. Aortic cells from the chimeras were analysed using mass cytometry and 8 myeloid-cell populations were identified (Fig. 6b). Each cell population was divided into CD45.1⁺ (host), CD45.1⁺CD45.2⁺ (WT donor), and CD45.2⁺ (*Clec4a2*$^{−/−}$ donor), and the proportion of donor origins in each population was calculated. In line with the results from the in vitro competitive culture with CSF1 or CSF2 (Fig. 5e,g), WT-derived BM cells outcompeted *Clec4a2*$^{−/−}$ cells in the resident macrophage pool in the healthy aorta (Fig. 6c,d), whereas *Clec4a2*$^{−/−}$ BM cells preferentially gave rise to either Ly6C⁺ monocytes or cDC2 (Fig. 6e,f).

A similar reconstitution of myeloid cells by WT and *Clec4a2*$^{−/−}$ cells was observed in the atherosclerotic aorta (Fig. 6g). Cells deficient in CLEC4A2 displayed a significantly reduced capacity to replenish the vascular resident macrophage pool compared with WT cells (Fig. 6h), but preferentially accumulated as Ly6C⁺ monocytes, cDC2, and CD11c⁺ macrophages (Fig. 6i–k), the latter predominantly emerging during atherogenesis (Supplementary Fig. 1b). In summary, CLEC4A2 promotes the maintenance of the CSF1-dependent resident macrophage pool in the healthy and atherosclerotic artery, limiting the accumulation of pro-inflammatory myeloid subsets in atherogenesis.

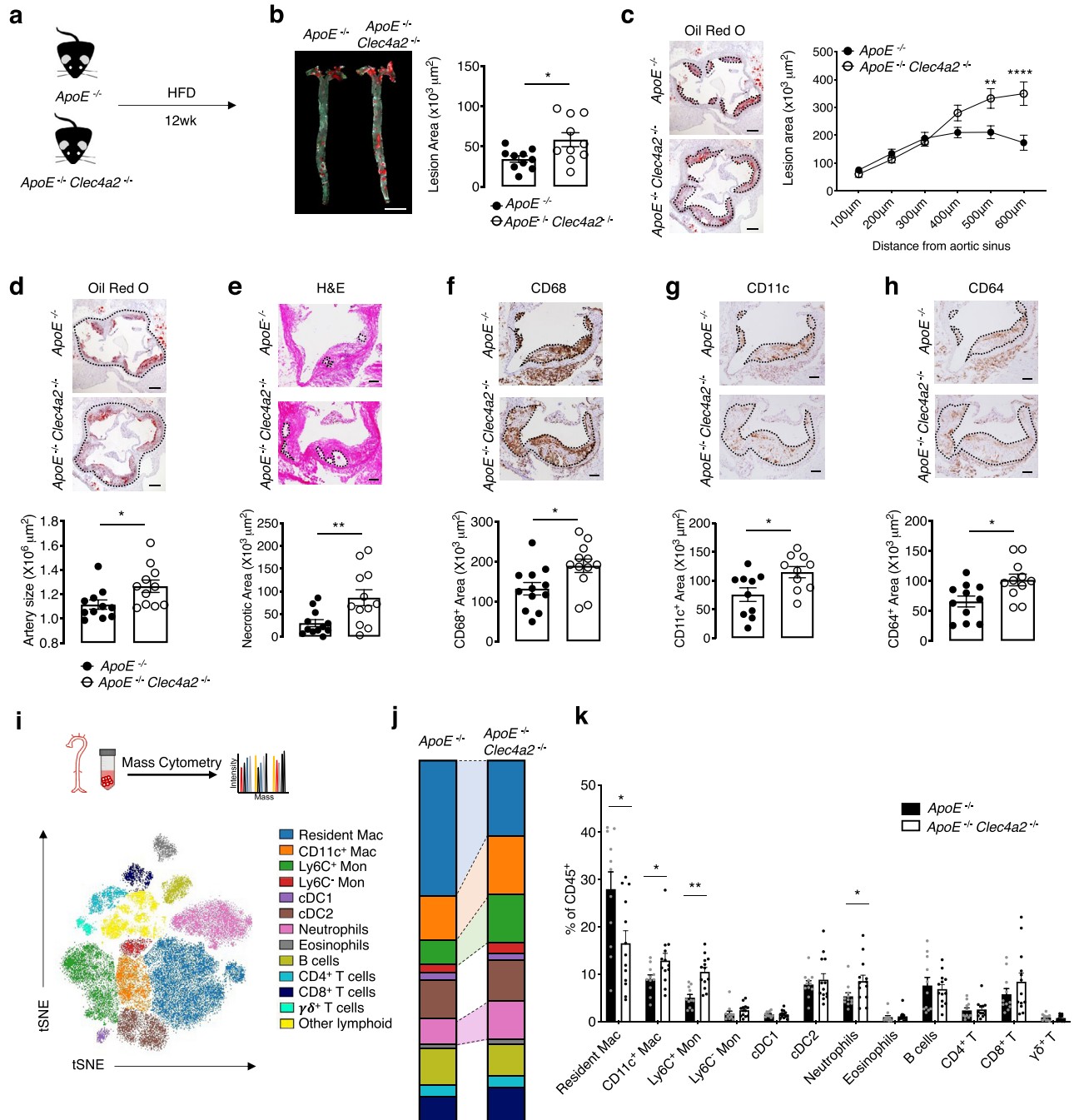

**Fig. 4 CLEC4A2 alleviates atherogenesis by maintaining vascular myeloid cell homeostasis. a** Schematic showing the experimental setup to compare 20-week-old $ApoE^{-/-}$ and $ApoE^{-/-}$ $Clec4a2^{-/-}$ littermate mice fed a HFD for 12 weeks. **b** Representative images and plaque quantification of Sudan IV (red)-stained en face whole aortas of HFD-fed $ApoE^{-/-}$ and $ApoE^{-/-}$ $Clec4a2^{-/-}$ littermate mice (Scale bar: 5 mm) ($n = 10$ mice each). Two-tailed Student's $t$-test. $*P = 0.0211$. **c** Representative images and plaque quantification of Oil Red O-stained aortic roots (Scale bar: 200 μm) ($n = 12$ mice each). Two-way ANOVA. $**P = 0.0025$, $****P < 0.0001$. **d** Representative images of aortic roots showing arterial size (dotted area) (Scale bar: 200 μm) ($n = 11$ mice each). Two-tailed Student's $t$-test. $*P = 0.0279$. **e** Representative images of the aortic root lesion stained with hematoxylin and eosin (H&E) indicating the necrotic core (H&E-free area, dotted) (Scale bar: 100 μm) ($n = 12$ mice each). Two-tailed Student's $t$-test. $**P = 0.0080$. **f–h**, Representative images and quantification of CD68 (**f**), CD11c (**g**), and CD64 (**h**)-stained (brown)-area in aortic root lesions (dotted). (Scale bar: 100 μm). Two-tailed Student's $t$-test. CD68: $n = 12$ mice each ($*P = 0.0201$), CD11c: $n = 9$–10 mice each ($*P = 0.0205$), CD64: $n = 11$ mice each ($*P = 0.0137$). **i**, tSNE clustering of live CD45$^+$ cells from whole aortas of $ApoE^{-/-}$ and $ApoE^{-/-}$ $Clec4a2^{-/-}$ mice fed a HFD for 12 weeks by mass cytometry. Leucocyte populations were identified based on expression of 32 markers. **j** and **k** Frequency of leucocyte populations in each genotype ($ApoE^{-/-}$: $n = 11$ mice; $ApoE^{-/-}$ $Clec4a2^{-/-}$: $n = 13$ mice, pooled from three independent experiments). Two-tailed Student's $t$-test. Resident Mac: $*P = 0.0165$, CD11c$^+$ Mac: $*P = 0.0477$, $**$Ly6C$^+$ Mon: $**P = 0.0084$, neutrophils: $*P = 0.0469$. All data are presented as mean ± SEM. Source data are provided as a Source Data file.

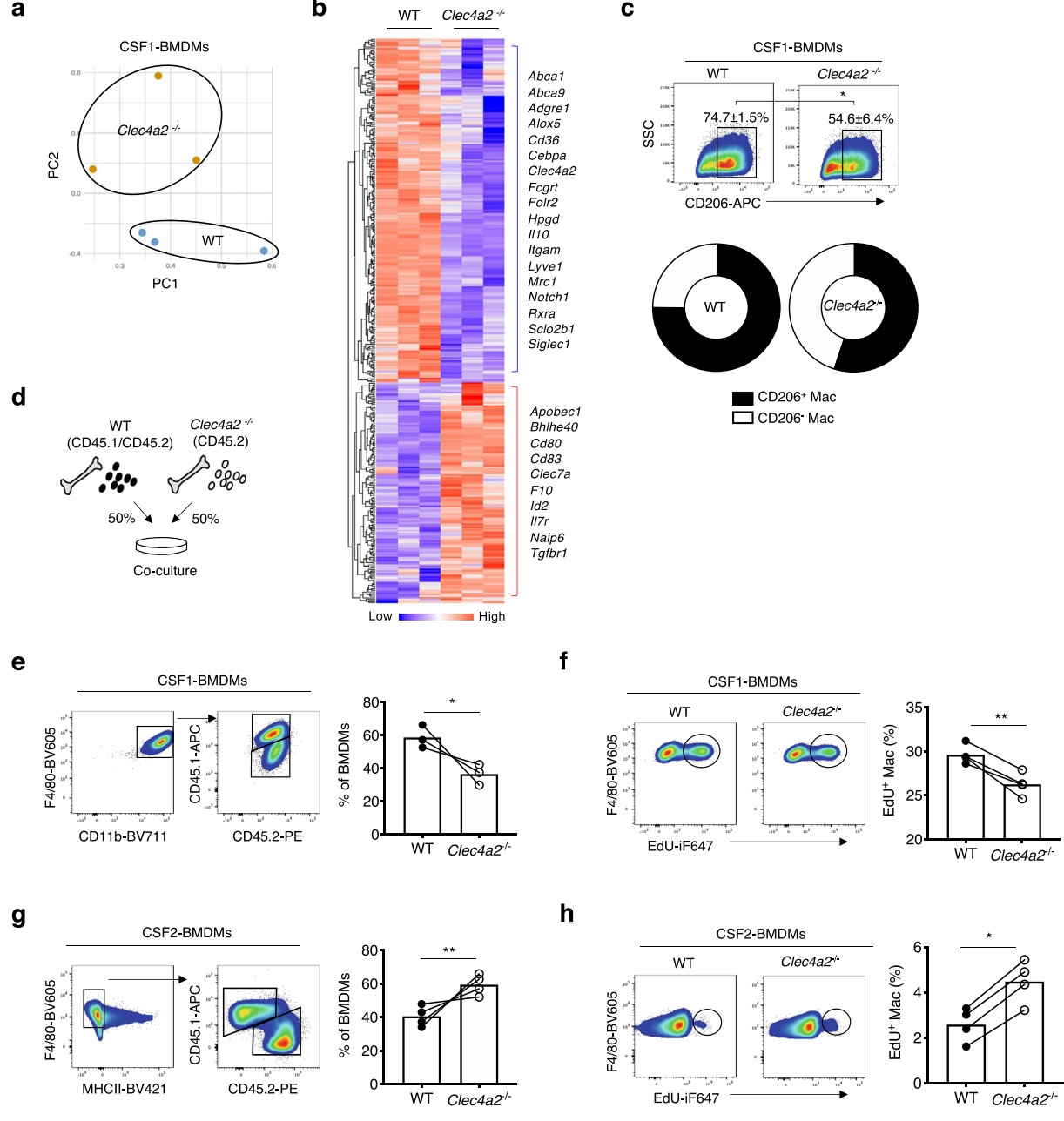

**Fig. 5 CLEC4A2 maintains macrophage identity and homeostasis by promoting CSF1- over CSF2-dependent differentiation in vitro. a** Principal component analysis (PCA) of the RNA-seq data from CSF1-BMDMs of WT and *Clec4a2*−/− mice. **b** Heatmap of differentially expressed genes (*P* < 0.05) between WT and *Clec4a2*−/− CSF1-BMDMs by RNA-seq (*n* = 3 mice, the heatmap represents one of two independent experiments). Differential expression was assessed using the two-sided Robinson and Smyth exact test with Bonferroni's correction for multiple-hypothesis testing. **c** Expression of CD206 in WT and *Clec4a2*−/− CSF1-BMDMs by flow cytometry (*n* = 5 mice each; the data are representative of two independent experiments). Two-tailed Student's *t*-test. *P = 0.0158. **d** Schematic showing coculture of BM cells from 50% WT (CD45.1/CD45.2 heterozygous) and 50% *Clec4a2*−/− (CD45.2) mice with CSF1. **e** Representative flow cytometry plots and proportion of each origin in CD11b+F4/80+CD206+ CSF1-BMDMs (*n* = 3 mice each; the data are representative of two independent experiments). Two-tailed Student's *t*-test. *P = 0.0148. **f** Representative flow cytometry plots and percentages of 5-ethynyl-2′-deoxyuridine (EdU) incorporation in cocultured BMDMs (CD11b+F4/80+CD206+) with CSF1 (*n* = 3 mice each; one experiment). Two-tailed Student's *t*-test. **P = 0.0048. **g** BM cells of 50% WT (CD45.1) and 50% *Clec4a2*−/− (CD45.2) mice were cocultured with CSF2. Adherent cells were analysed by flow cytometry. Representative flow cytometry plots and proportion of each origin in CD11b+CD11c+F4/80+MHCII− CSF2-BMDMs (*n* = 4 mice each; the data are representative of two independent experiments). Two-tailed Student's *t*-test. **P = 0.0083. **h** Representative flow cytometry plots and percentages of EdU incorporation in cocultured BMDMs (CD11b+CD11c+F4/80+MHCII−) with CSF2 (*n* = 3 mice each, one experiment). Two-tailed Student's *t*-test. *P = 0.0208. All data are presented as mean ± SEM. Source data are provided as a Source Data file.

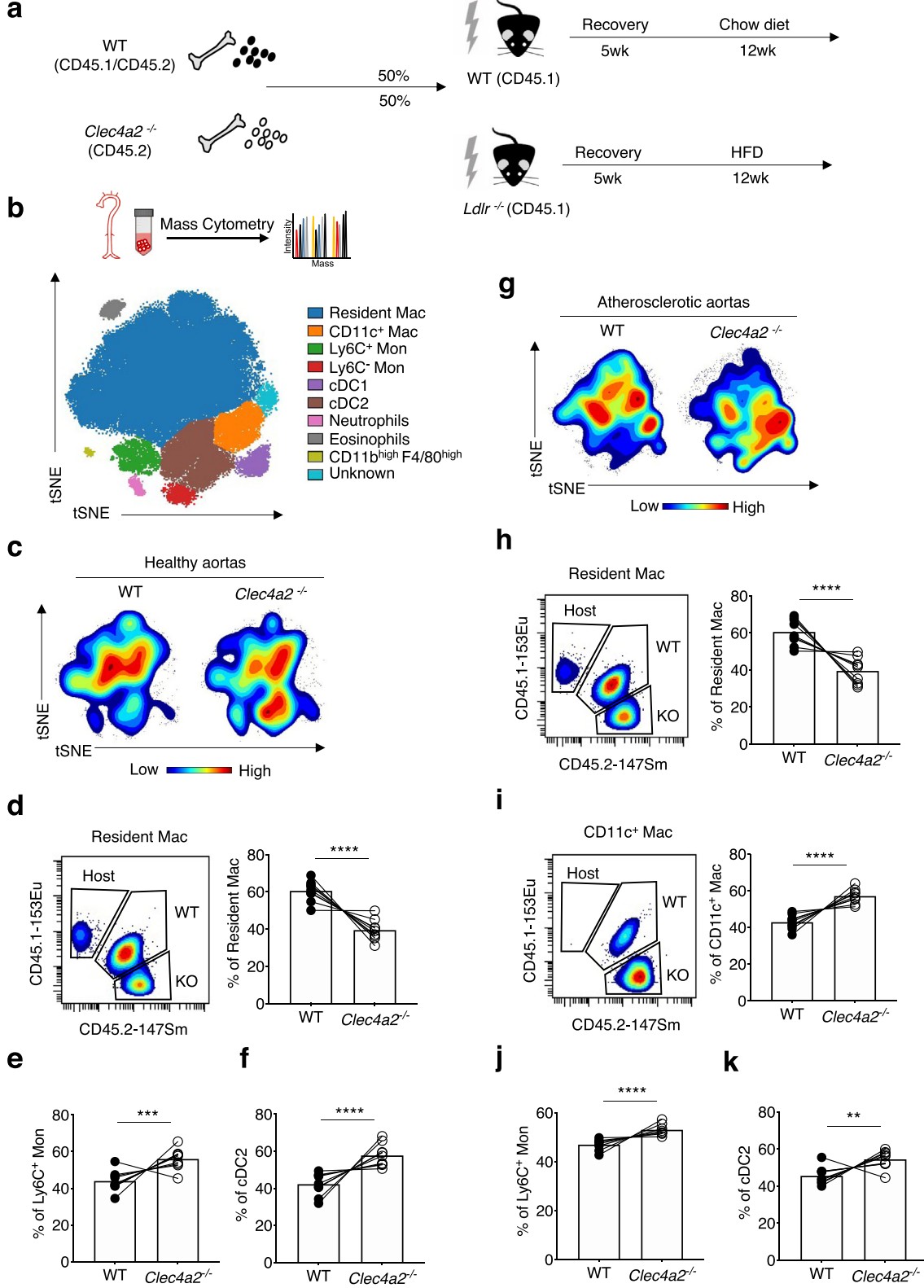

We then sought to investigate whether the proliferation capacity of vascular macrophages was affected by CLEC4A2 deficiency by injecting BrdU in the steady state and in atherosclerotic mice. We detected that there were no apparent differences in BrdU incorporation in the resident macrophages from WT and $Clec4a2^{-/-}$ mice in the steady state (Supplementary Fig. 8a–c). However, differential effects of genetic deletion were observed in the atherosclerotic mice. $ApoE^{-/-}$ $Clec4a2^{-/-}$ mice compared with their $ApoE^{-/-}$ littermate mice showed a reduction in proliferation in resident macrophages and an increase in $CD11c^+$ macrophages (Supplementary Fig. 8d–h). These findings support our claim that CLEC4A2 drives loss of the resident macrophage pool while promoting lesional macrophage accumulation during atherosclerosis.

**Fig. 6 CLEC4A2 promotes tissue adaptation of vascular resident macrophages in the steady state and atherosclerosis. a** Schematic of competitive BM chimera experiments in the steady state and in atherosclerosis. Mixed BM (50 % WT (CD45.1 CD45.2 heterozygous) and 50% $Clec4a2^{-/-}$ (CD45.2)) cells were transplanted into WT (CD45.1) mice or $Ldlr^{-/-}$ (CD45.1) mice. The $Ldlr^{-/-}$ chimeras received a HFD for 12 weeks. Aortic cells were analysed by mass cytometry. **b** tSNE clustering of myeloid cells in aortas of WT and $Ldlr^{-/-}$ chimeric mice by mass cytometry. Myeloid populations were clustered based on the expression of 21 markers. Each population was divided by their origin using CD45.1 and CD45.2 marker expression. **c** Representative density plots of myeloid cells in the aorta of WT recipient mice by WT and $Clec4a2^{-/-}$ donor origins. ($n = 9$ mice each, pooled from three independent experiments). **d–f** Representative mass cytometry plots and percentages of resident macrophages, Ly6C$^+$ monocytes, and cDC2 from each origin in the aorta of the WT host. Two-tailed Student's $t$-test. Resident Mac: ****$P < 0.0001$, Ly6C$^+$ Mon: ***$P = 0.0003$, cDC2: ****$P < 0.0001$. **g** Representative density plots of myeloid cells in the aorta of HFD-fed $Ldlr^{-/-}$ mice by WT and $Clec4a2^{-/-}$ donor origins ($n = 9$, pooled from three independent experiments). **h–k** Representative mass cytometry plots and percentages of resident macrophages, CD11c$^+$ macrophages, Ly6C$^+$ monocytes and cDC2 from each origin in the aorta of the $Ldlr^{-/-}$ host. Two tailed Student's $t$-test. Resident Mac, CD11c$^+$ Mac and Ly6C$^+$ Mon: ****$P < 0.0001$, cDC2: **$P = 0.0011$. All data are presented as mean ± SEM. Source data are provided as a Source Data file.

**CLEC4A2 protects from atherosclerosis by controlling homeostatic and anti-atherogenic functions in vascular resident macrophages.** To understand how CLEC4A2 deficiency affects the vascular myeloid landscape during atherogenesis at the single-cell level, we performed scRNA-seq on live CD45$^+$CD11b$^+$ myeloid cells isolated from aortas of $ApoE^{-/-}$ and $ApoE^{-/-}$ $Clec4a2^{-/-}$ mice fed a HFD for 12 weeks. Clustering was performed using Seurat and similar myeloid cell populations were identified as described in Fig. 1 (Fig. 7a; Supplementary Data 4). scRNA-seq in Fig. 7b revealed a decrease in the resident macrophage cluster (C1) in $ApoE^{-/-}$ $Clec4a2^{-/-}$ mice, in line with the finding by mass cytometry (Fig. 4j). Clustering did not identify CD209b+Lyve1$^+$ macrophages as a separate cluster in this dataset (compared with Fig. 1), possibly due to the considerable decrease of the number of resident macrophages in $ApoE^{-/-}$ $Clec4a2^{-/-}$ mice. Compared with $ApoE^{-/-}$ mice, an increase in $Itgax$ (CD11c) expressing macrophage clusters (C2 and C6), proliferating macrophages (C8), Ly6C$^+$ monocytes (C5), and cDC2 (C7) was observed in the aorta of $ApoE^{-/-}$ $Clec4a2^{-/-}$ mice.

Since vascular resident macrophages expressed the highest level of $Clec4a2$ (Fig. 1c), we compared transcriptomic profiles of C1 from the two genotypes. CLEC4A2 deficiency resulted in downregulation of genes associated with macrophage identity ($Fcgr1$, $Cd68$, $Notch2$, $Timd4$, $Cd209g$, $Cd209f$, $Irf4$, and $Scarb1$) (Fig. 7c; Supplementary Data 5). Importantly in relation to atherosclerosis, genes related to lipid metabolism, which featured in resident macrophages (Fig. 1e), were downregulated ($Plin2$, $Trem2$, $Lipa$, $Msr1$, and $Abcg1$) by CLEC4A2 deletion. In contrast, genes associated with inflammation ($Tlr2$, $Il1b$, $Il6$, $Stat5a$, and $Il1a$) were upregulated in C1 from $ApoE^{-/-}$ $Clec4a2^{-/-}$ mice. When comparing bulk and scRNA-seq, we found 48 conserved genes in $Clec4a2^{-/-}$ C1-resident macrophages (Fig. 7c) and $Clec4a2^{-/-}$ CSF1-BMDMs (Fig. 5a,b), with 31 genes downregulated and 17 genes upregulated (Supplementary Table 6).

To verify the functional significance of these transcriptomic changes in CLEC4A2-deficient macrophages, we tested the ability of $Clec4a2^{-/-}$ CSF1-BMDMs to respond to toll-like receptor (TLR) ligands. $Clec4a2^{-/-}$ macrophages produced higher levels of the pro-inflammatory cytokine IL-6 when stimulated by FSL-1 (TLR2 ligand) or LPS (TLR4 ligand) compared with the WT cells (Fig. 7d,e), demonstrating that CLEC4A2 inhibits TLR signalling in macrophages.

Second, we tested how CLEC4A2 modulates the ability of macrophages to uptake and process oxidised LDL (oxLDL). CSF1-BMDMs from WT and $Clec4a2^{-/-}$ mice were treated with oxLDL for 24 h. Greater lipid levels accumulated in the cytoplasm of $Clec4a2^{-/-}$ BMDMs compared with the WT counterpart (Fig. 7f,g). We then examined the reverse cholesterol-transport capacity, and found that upon oxLDL loading $Clec4a2^{-/-}$ macrophages were less capable of processing and secreting cholesterol than WT cells (Fig. 7h). Consistently, oxLDL-laden

$Clec4a2^{-/-}$ macrophages showed lower expression of cholesterol-efflux genes ($Abca1$, $Abcg1$) relative to the WT cells (Fig. 7i). Since CLEC4A2 signals through its ITIM with phosphatase SHP1, we tested whether SHP1 inhibition can phenocopy the effect of CLEC4A2 deficiency on oxLDL uptake. Suppression of SHP1 activity increased lipid accumulation in WT BMDMs but not in $Clec4a2^{-/-}$ macrophages (Fig. 7j) and reduced cholesterol efflux genes in WT BMDMs (Fig. 7k). Our results demonstrate that CLEC4A2 prevents pro-atherogenic macrophage behaviour by promoting cholesterol handling in a SHP1-modulated manner.

## Discussion

In our study, we show that the C-type lectin receptor CLEC4A2 identifies a subpopulation of Lyve1$^+$ vascular resident macrophages endowed with athero-protective functions (Figs. 1, 2, 3). CLEC4A2 licences monocyte differentiation to vascular resident macrophages and their acquisition of homeostatic properties by promoting CSF1-driven over CSF2-driven monocyte-fate decisions and proliferation, avoiding losses of the vascular resident macrophage pool (Figs. 5, 6, Supplementary Fig. 8). During atherogenesis, CLEC4A2 limits TLR signalling and promotes cholesterol efflux in vascular macrophages (Fig. 7), which in turn protects from atherogenesis (Fig. 4). Our findings advance the understanding of the mechanisms of macrophage tissue adaptation to the vascular niche through CLEC4A2 and highlight the role of CLEC4A2$^+$ macrophages in atheroprotection.

Vascular macrophages are heterogeneous and the contribution of each subset to atherogenesis remains elusive. By using mass cytometry, we show that the intima and adventitia of the atherosclerotic aorta have a different immune landscape with two distinct macrophage compartments (Supplementary Fig. 1). Our scRNA-seq data on myeloid cells in murine atherosclerotic aortas provide a high-resolution landscape of the myeloid lineage, identifying 12 cell populations including 8 macrophage clusters (Fig. 1). These macrophage subsets include previously identified clusters such as TREM2$^{high}$ [11], inflammatory[13], IFN-inducible[15], and proliferating[15]. Our analysis uncovers significant heterogeneity in $Lyve1^+Mrc1^+Pf4^+$ vascular resident macrophages, segregating them into 3 subsets. These comprise two subsets that share a homeostatic gene signature (cholesterol handling, vascular cell maintenance, phagocytosis, and wound healing) and one with an activated resident signature (responses to wounding, chemotaxis, responses to oxidised phospholipids, NLRP3 inflammasome, and NFκB pathways). Importantly, $Clec4a2$ is one of the top identifier genes discriminating homeostatic from activated $Lyve1^+Mrc1^+Pf4^+$ resident macrophages in the atherosclerotic aorta. Ablation of CLEC4A2$^+$-resident macrophages using LysM$^{Cre+}$ $Clec4a2^{flox/DTR}$ mice increases lesion development during atherogenesis (Figs. 2, 3), indicating that CLEC4A2$^+$ macrophages are a subpopulation of Lyve1$^+$ vascular resident macrophages endowed with protective functions as predicted by their gene expression profile in scRNA-seq (Fig. 1).

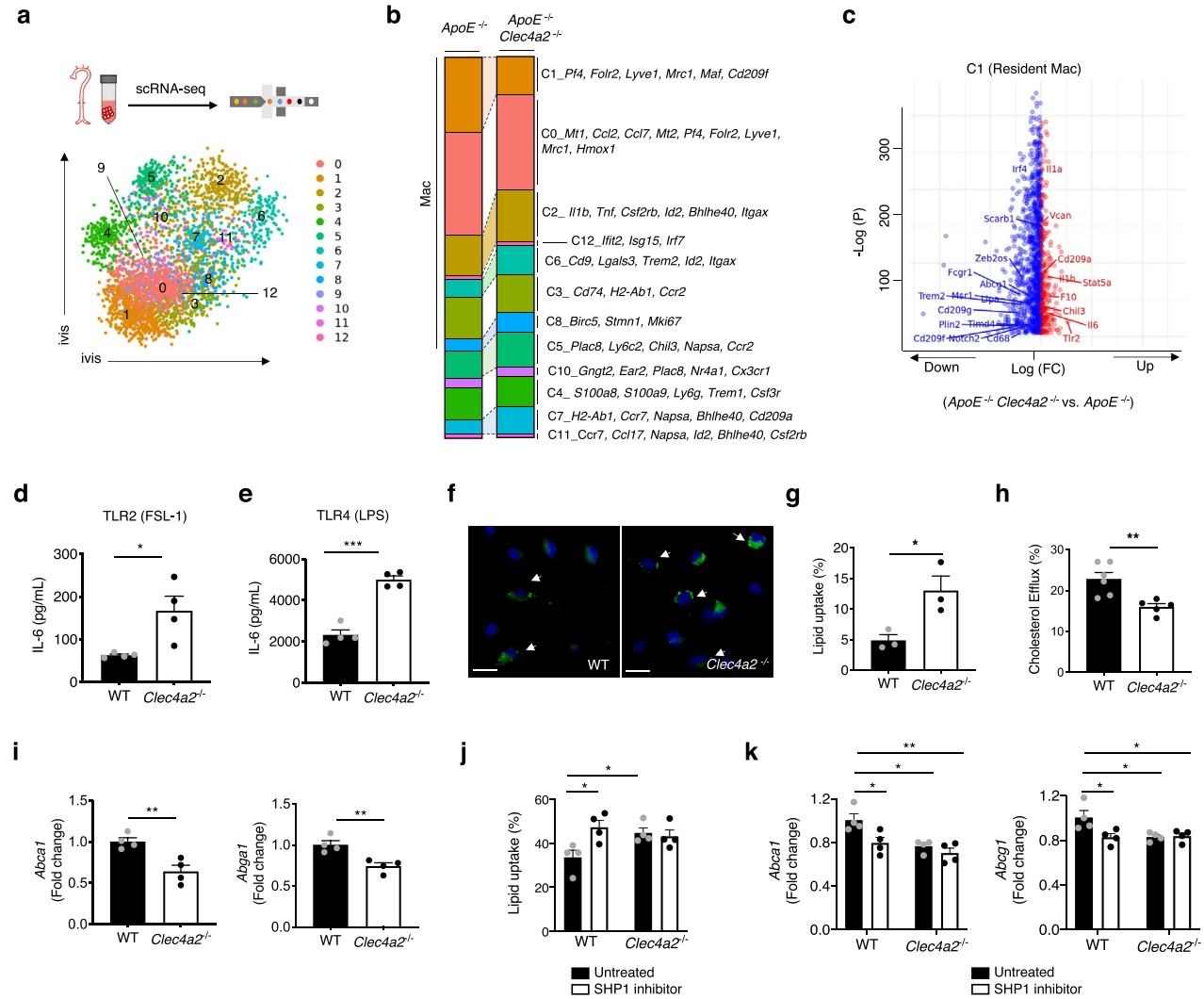

**Fig. 7 CLEC4A2 protects from atherosclerosis by promoting homeostatic functions in macrophages. a** CD45⁺CD11b⁺ cells from whole aortas of *ApoE*⁻/⁻ and *ApoE*⁻/⁻ *Clec4a2*⁻/⁻ mice fed a HFD for 12 weeks were analysed by scRNA-seq using 10X genomics platform. Aortic cells were pooled from 9 mice per genotype group. ivis clustering of myeloid cells pooled from 18 mice showed 13 clusters. **b** Proportion of 12 myeloid clusters in each genotype. **c** Volcano plot comparing gene expression (*P* < 0.05) in *ApoE*⁻/⁻ *Clec4a2*⁻/⁻ Cluster 1 (resident macrophages) vs. *ApoE*⁻/⁻ Cluster 1. Differential expression was assessed using the two-sided Robinson and Smyth exact test with Bonferroni's correction for multiple-hypothesis testing. **d** and **e** IL-6 production by CSF1-BMDMs from WT and *Clec4a2*⁻/⁻ mice in response to FSL-1 (a ligand for TLR2) or LPS (a ligand for TLR4) treatment for 24 h, *n* = 4 mice each; the data are representative of four independent experiments. Two-tailed Student's *t*-test. *\**P* = 0.0224. ***\**P* = 0.0001. **f** and **g** Representative images of oxidised low-density lipoprotein (oxLDL) uptake in CSF1-BMDMs from WT and *Clec4a2*⁻/⁻ mice in 24 h and percentages of oxLDL-laden BMDMs (Scale bar: 10 μm), *n* = 3 mice each; the data are representative of three independent experiments. Two-tailed Student's *t*-test. *\**P* = 0.0331. **h** The efficiency of reverse cholesterol efflux in oxLDL-laden BMDMs from WT and *Clec4a2*⁻/⁻ mice. *n* = 5-6 mice each; the data are representative of three independent experiments. Two-tailed Student's *t*-test. *\**P* = 0.0055. **i** Gene expression of cholesterol efflux genes (*Abca1* and *Abga1*) in WT and *Clec4a2*⁻/⁻ BMDMs treated with oxLDL for 24 h. The expression levels were normalised against *Actb*. *n* = 4 mice each; the data are representative of two independent experiments. Two-tailed Student's *t*-test. *Abca1*: *\**P* = 0.0061, *Abga1*: *\**P* = 0.0065. **j** The effect of SHP1 inhibition on oxLDL uptake in CSF1-BMDMs from WT and *Clec4a2*⁻/⁻ mice, *n* = 4 mice each, one experiment. Ordinary one-way ANOVA with Holm–Sidak's multiple-comparison test. UT (WT) vs. inhibitor (WT): *\**P* = 0.0185; UT (WT) vs. UT (*Clec4a2*⁻/⁻): *\**P* = 0.0388. **k** Gene expression of *abca1* and *abga1* in WT and *Clec4a2*⁻/⁻ BMDMs treated with oxLDL in the absence or presence of SHP1 inhibitor for 48 h. The expression levels were normalised against *Actb* (*n* = 4 mice each; one experiment). Ordinary one-way ANOVA with Holm–Sidak's multiple-comparison test. *Abca1*: UT (WT) vs. inhibitor (WT): *\**P* = 0.0314, UT (WT) vs. UT (*Clec4a2*⁻/⁻): *\**P* = 0.0171, UT (WT) vs. inhibitor (*Clec4a2*⁻/⁻): *\**P* = 0.0038. *Abga1*: UT (WT) vs. inhibitor (WT): *\**P* = 0.0421, UT (WT) vs. UT (*Clec4a2*⁻/⁻): *\**P* = 0.0421, UT (WT) vs. inhibitor (*Clec4a2*⁻/⁻): *\**P* = 0.0421. All data are presented as mean ± SEM. Source data are provided as a Source Data file.

Although CLEC4A2 expression identifies macrophages with specific properties, its expression is not exclusive to vascular macrophages[29]. CLEC4A2 is highly expressed by alveolar and cardiac macrophages (Supplementary Fig. 3), and they are also ablated by DT in *LysM*^Cre+ *Clec4a2*^flox/DTR mice (Supplementary Fig. 5). Nonetheless, there is no evidence to date that these

macrophages have a direct impact on atherogenesis. In addition, neutrophils express CLEC4A2 in the BM and blood, but their CLEC4A2 expression is low in the artery. We observed a transient depletion of neutrophils in the BM after DT administration to *LysM*^Cre+ *Clec4a2*^flox/DTR mice, with full recovery at 48 h (Supplementary Fig. 5d). DT-mediated depletion did not affect

neutrophils in peripheral tissues such as spleen and aorta (Fig. 2; Supplementary Fig. 5). Ablation of neutrophils has been shown to reduce the progression of atherogenesis[30]. Therefore, the effect of short-term neutrophil depletion in the BM may contribute to underestimate the pro-atherogenic effect of ablation of CLEC4A2+ macrophages. Finally, DT-mediated cell ablation did not induce systemic inflammation or increased serum cholesterol levels (Supplementary Table 2). Thus, the pro-atherogenic effects of DT-mediated cell ablation are on the whole driven by ablation of CLEC4A2+ macrophages.

Our findings show that CLEC4A2 has an essential role in deciding the fate of monocytes as they differentiate in situ (Figs. 5, 6). CSF1 induces housekeeping features that characterise tissue resident macrophages, while CSF2-dependent macrophages acquire pro-inflammatory functions[31]. The fine balance between CSF1 and CSF2 signalling is important in tissue homeostasis, for instance, by preserving the protective function of the microglia[32]. Vascular resident macrophages in the steady state are CSF1-dependent[6,7] while CSF2 underpins the survival and activation of intimal CD11c+ myeloid cells in atherosclerosis[33,34]. We provide in vitro (Fig. 5) and in vivo (Fig. 6; Supplementary Fig. 8) evidence that CLEC4A2 biases the monocytes to macrophage transition in response to CSF1 by monitoring proliferation and differentiation in WT and $Clec4a2^{-/-}$ cells. As a result of this bias, CLEC4A2-competent monocytic cells are more apt to differentiate into CSF1-dependent resident macrophages than CLEC4A2-deficient ones, and $Clec4a2^{-/-}$ macrophages remain monocytes or differentiate into CSF2-dependent cell states such as cDC2 and CD11c+ macrophages (Figs. 5 and 6). Our in vivo data recapitulate in vitro cellular phenotypes as shown by the shared genes regulated by CLEC4A2 in macrophages collected from the aorta and in culture as assessed by scRNA-seq and bulk RNA-seq (Supplementary Table 6). Our RNA-seq data demonstrate that CLEC4A2 deletion induces a differentiation bias by causing profound transcriptional changes, with loss of core macrophage genes (e.g., $Mrc1$, $Siglec1$, $Slco2b1$, $Folr2$, $Adgre1$, $Fcgrt$, and $Lyve1$ in BMDMs; $Fcgr1$, $Cd209g$ and $Timd4$ in vascular resident macrophages) and regulators of macrophage maturation ($Notch1$, $Cebpa$, and $Irf4$ in BMDMs; $Notch2$, $Zeb2os$, and $Irf4$ in vascular resident macrophages) (Figs. 5, 7). Notch signalling is required for appropriate differentiation of macrophages that perform tissue repair[35] and contributes to maturation of Kupffer cells[36,37]. Interferon regulatory factor 4 (IRF4) is a key transcription factor in polarisation of alternatively activated macrophages[38]. ZEB2os is a long noncoding RNA that regulates the translation of ZEB2[39]. Loss of macrophage identity by ZEB2 deletion results in diminished tissue-macrophage pools in the liver and lung[40]. Downregulation of these genes in CLEC4A2-deficient macrophages coincides with the enhanced expression of CSF2-dependent transcription factors, $Id2^{41}$ and $Bhlhe40^{42}$ (Fig. 5). These genes are consistently identifiable in CD11c+ macrophage subsets in the aorta ($Id2$ in inflammatory macrophages (C2) and TREM2high macrophages (C6), and $Bhlhe40$ in C2; Fig. 7). Both subsets are increased by CLEC4A2 deficiency during atherosclerosis (Fig. 7). Together, functional and transcriptional findings indicate that CLEC4A2-dependent changes in macrophages are ascribable to a single overarching mechanism of bias in monocyte-fate decision from CSF2 to CSF1 within the arterial wall. Future studies will help mine further the mechanisms underlying CSF1/CSF2-regulated events in arterial macrophages.

CLEC4A2 has an atheroprotective role and its genetic deletion causes an increase in lesion size, accumulation of inflammatory myeloid cells, and necrotic core formation (Fig. 4). We show that CLEC4A2 has an important role in minimising losses of the vascular resident macrophage pool during atherogenesis. A concomitant

dilatation of the vessel wall, consistent with the finding by Lim et al.[7], also further supports this conclusion. The lack of systemic increase in myeloid subsets in the BM, blood, spleen and para-aortic lymph nodes in $Clec4a2$-deficient atherosclerotic mice (Supplementary Fig. 7), indicates that CLEC4A2 protects from atherogenesis through its effects within the arterial wall. Our findings in a competitive BM chimera setting indicate that the cell-intrinsic effect of CLEC4A2 on differentiation is driving the loss of the vascular resident macrophage pool (Fig. 6). The rise in CD11c+ macrophages by CLEC4A2 deficiency is observed when the lesional macrophage niche is expanded in atherosclerosis (Fig. 4 and Supplementary Fig. 7). Notably, the loss of the vascular resident macrophage pool in the absence of CLEC4A2 precedes the changes in lesion formation and myeloid landscape (Fig. 4), demonstrating that the primary function of CLEC4A2 is to preserve the size of the vascular resident macrophage pool during atherogenesis. These findings, together with the evidence of a pro-atherogenic effect of CLEC4A2+ macrophage depletion (Fig. 3), reveal that CLEC4A2 is essential in preventing loss of the vascular resident macrophage pool by promoting monocyte differentiation into homeostatic vascular resident macrophages and thus protecting the aorta from atherosclerosis.

Macrophages are integral to inflammation and cholesterol homeostasis, key biological processes in atherogenesis. CLEC4A2 not only controls the vascular resident macrophage pool size, but also ensures their appropriate functional tissue adaptation. Using a combination of scRNA-seq and functional assays, we reveal that CLEC4A2-dependent macrophage programming has two key functional consequences that protect from atherogenesis. The first is its ability to dramatically reduce TLR2 and TLR4 signalling in macrophages (Fig. 7). Resident macrophages restrain their TLR responses to nucleic acids when clearing cellular debris[43]. Human CLEC4A and murine CLEC4A4, an isoform of CLEC4A2 bearing the ITIM, hamper TLR8[44] or TLR2/3/4/9 responses in DCs[45]. SHP1 can act as a negative regulator of TLR signalling[46,47] and the deletion of the TLR/IL-1R adaptor MyD88 in $motheaten$ mice rescues its hyperinflammatory phenotype[48]. We also find that in the absence of CLEC4A2, vascular resident macrophages upregulate inflammatory gene expression ($Il1b$, $Tlr2$, and $Stat5a$) and BMDMs display enhanced responses when exposed to TLR2 and TLR4 ligands. In atherosclerosis, lipoproteins trigger TLR2 and TLR4 signalling pathways[49,50]. TLR2 and TLR4 are pro-atherogenic and enhance plaque inflammation via the production of cytokines and activation of the inflammasome[51,52]. TLR activation is also key in driving changes in macrophages in the vessel wall. For instance, TLR4-mediated IRF5 activation increases CD11c expression in vascular macrophages[53]. Our functional and transcriptional data indicate that CLEC4A2 desensitises macrophages from continuous TLR triggering during atherogenesis, protecting from excessive inflammation. It is notable that $Clec4a2$ expression in macrophages is downregulated upon inflammatory stimuli (Supplementary Fig. 3), suggesting that CLEC4A2 may act as a switch to allow sensitisation of macrophages to TLR signalling when inflammation is detected in the vessel wall.

Last, our scRNA-seq data show that vascular resident macrophages have a transcriptional signature of genes involved in cholesterol efflux (Fig. 1), which is essential for appropriate lipid metabolism within the vessel wall[54]. Loss of CLEC4A2 interferes with the expression and function of reverse cholesterol-transport pathways. In the absence of CLEC4A2, macrophages express lower levels of essential cholesterol-efflux genes ($Abca1$ and $Abcg1$) (Figs. 5, 7), and are consequently less capable of performing the reverse cholesterol transport in response to oxLDL loading, resulting in accumulation of lipids and foam-cell formation (Fig. 7). CLEC4A2 is one of the few CLRs bearing a cytoplasmic ITIM that recruits SHP1 and SHP2[18,55]. Both SHP1-

deficient (*motheaten*) and CLEC4A2-deficient mice are characterised by DC expansion, enhanced TLR and CSF2 signalling, and susceptibility to autoimmune diseases[21,56,57]. While the role of the CLEC4A2 and SHP1 in TLR and CSF2 signalling is well documented[21,56–58], whether these pathways modulate cholesterol handling is unknown. We demonstrate that intracellular cholesterol accumulation and *Abca1* and *Abcg1* expression is modulated by SHP1, directly linking activation of the CLEC4A2 signalling pathway and atheroprotection. Our study indicates that CLEC4A2 is a crucial regulator of vascular-macrophage functions at the crossroads of immune and cholesterol homeostasis in atherosclerosis.

The findings of our work challenge the widely held belief that macrophages are solely detrimental in cardiovascular disease with the discovery of the protective subset of CLEC4A2+ vascular macrophages. Our data identify CLEC4A2 as a crucial receptor for tissue adaptation of macrophages, which has a protective role in vascular health and disease. Guided by single-cell biology, our data pave the way to discoveries on the fine functional heterogeneity of vascular macrophages and how we modulate specific subsets to alter the course of atherogenesis. Fostering the homeostatic functions of macrophages through the CLEC4A pathway might represent a transformative therapeutic strategy for inflammatory and cardiovascular diseases.

## Methods

**Animals.** All experimental animal procedures were approved by The University of Oxford Animal Welfare Ethical Review Board and performed according to UK Home Office regulations and The University of Oxford guidelines. UK Home Office regulations conform to the guidelines from Directive 2010/63/EU of the European Parliament on the protection of animals used for scientific purposes. Mice were maintained in individually ventilated cages at 20–23.3 °C under a 12-hour light/dark cycle with free access to food and water. Mice were fed a chow diet (Special Diets Services; Standard RM3 (P) 801700) unless stated otherwise. All mouse strains were on a C57BL/6 J background and bred under specific pathogen-free conditions. Mouse strains used for experiments include C57BL/6 J wild type (WT) (The Jackson Laboratory, Stock No. JAX:000664), *ApoE*−/− mice (The Jackson Laboratory, Stock No. JAX:002052), *Ldlr*−/− (The Jackson Laboratory, Stock No. JAX: 002207), *LysM*Cre (The Jackson Laboratory, Stock No. JAX: 004781), CD45.1 (The Jackson Laboratory, Stock No. JAX: 002014), and *Cx3cr1*eGFP (The Jackson Laboratory, Stock No. JAX: 005582). Both male and female mice were used for all experiments and were age- and sex-matched, except for the RNA-seq experiments (Fig. 1, Fig. 5, and Fig. 7) and in vivo proliferation experiments (Supplementary Fig. 8) where only female mice were used.

Frozen embryos of the *Clec4a2*−/− mouse stain were obtained from Professor Yoichiro Iwakura (University of Tokyo, Japan)[21]. *Clec4a2*−/− mice were crossed with *ApoE*−/− mice to generate *ApoE*−/− *Clec4a2*−/− littermate mice. The *Clec4a2*flox/DTR mouse strain was designed by Monaco Lab and generated by Gen0way (France). A construct containing *loxP* sequences, neomycin, *Hbegf* (diphtheria-toxin receptor, DTR), and *egfp* was inserted upstream of exon 8 of the *Clec4a2* gene without affecting its transcription. *Clec4a2*flox/DTR homozygous mice were further crossed with *LysM*Cre homozygous mice. *Clec4a2*flox/DTR mice expressing *LysM*Cre (both heterozygous and homozygous) were tested in this study. *Ldlr*−/− CD45.1 mice were generated by crossing *Ldlr*−/− and CD45.1 mice.

**High-fat feeding.** In total, 8–10-week-old *ApoE*−/−, *ApoE*−/− *Clec4a2*−/−, and *Ldlr*−/− CD45.1 mice were fed a high-fat diet (HFD) *ad libitum* for 12–16 weeks and were examined at 20–24 weeks of age. A HFD was purchased from Special Diet Services (UK), consisting of (w/w) cocoa butter (15%), cholesterol (0.25%), maize starch (10%), casein (20%), sucrose (40.5%), cellulose (5.95%), corn oil (1%), 50% choline chloride (2%), methionine (0.2%), and mineral mixture (5.1%).

**Diphtheria-toxin-mediated cell ablation.** Diphtheria toxin (DT) (Sigma-Aldrich) was reconstituted in saline at 1 mg/ml. *Clec4a2*flox/DTR, *LysM*Cre+ *Clec4a2*flox/DTR, CD45.1 chimeric or *Ldlr*−/− CD45.1 chimeric mice received 30 ng/g DT intraperitoneally.

**Bone marrow chimeras.** In total, 8–10-week-old CD45.1 or *Ldlr*−/− CD45.1 mice were sub-lethally irradiated twice with a 4-hour interval using an X-ray irradiator (Gulmay; 1.8 Gy/min, 210 KV, 13 mA for 3 min). The irradiated mice received 3–5 × 10⁶ bone marrow (BM) cells by tail vein injection 1 h after the last irradiation. 1 % antibiotic (Baytril, Bayer) in autoclaved drinking water was given for 2 weeks.

**Adoptive transfer of Ly6C+ monocytes.** Monocytes were negatively isolated from the BM of 10-12-week-old *Cx3cr1*eGFP CD45.1 CD45.2 mice using Monocyte Isolation Kit (Miltenyi Biotec). 0.5–1 × 10⁶ monocytes in 100 μl were intravenously injected in 10-12-week-old *LysM*Cre+ *Clec4a2*flox/DTR or 26-28-week-old *Ldlr*−/− CD45.1 chimeric mice in two consecutive days.

**BrdU-incorporation assays.** In total, 12-week-old WT and *Clec4a2*−/− or 20-week-old HFD-fed *ApoE*−/− and *ApoE*−/− *Clec4a2*−/− mice received 5 intraperitoneal injections of 1 mg of BrdU over 9 days[8]. Aortic cells were stained using BrdU Flow kit (APC) (BD Biosciences), according to the manufacturer's instructions. For a macrophage depletion experiment, CD45.1 chimeric mice received 2 intraperitoneal injections of 1 mg of BrdU in two consecutive days post DT treatment.

**Tissue preparation.** All enzymes were purchased from Sigma-Aldrich, unless stated otherwise. Tissues were isolated in 5% FBS (Gibco) in RPMI-1640 (Gibco). Tissues were cut into small pieces and digested in the appropriate enzyme mixture at 37 °C in a water-bath shaker: Aorta:[59] 450 U/ml collagenase I, 125 U/ml collagenase XI 60 U/ml hyaluronidase and 60 U/ml DNase I (ThermoFisher Scientific) for 50 min; Lung: 0.4 mg/ml collagenase IV and 0.15 mg/ml DNase I for 40 min; Heart: 1 mg/ml collagenase II and 0.15 mg/ml DNase I for 30 min. The cells were retrieved by passing tissue pieces through a 70 μm cell strainer (Greiner Bio-One). Spleen and lymph nodes were mechanically dissociated by passing cells through a 70 μm cell strainer. For separating aortic layers into the intima/media and the adventitia, the whole aorta was pre-incubated with 125 U/ml collagenase II and 3.75 U/ml elastase at 37 °C for 10 min. The adventitia layer was separated from the inner tube (the intima and media) by pulling with two pairs of fine forceps. The tissues were then digested as performed in the aorta sample. For lungs, hearts, and spleens, erythrocytes were lysed using Red Cell Lysis Buffer (Sigma-Aldrich) at room temperature for 2 min.

**Flow cytometry.** Cells were stained with Live/Dead fixable aqua dye (Thermo-Fisher Scientific) for 10 min at room temperature and were then washed with 1% FBS in PBS (Lonza). Nonspecific binding of antibodies was blocked by incubating cells with rat anti-mouse CD16/32 (BD Biosciences) at 4 °C for 10 min, followed by incubating with fluorochrome-conjugated antibodies recognising cell-surface markers for 30 min at 4 °C. Following by two wash steps, cells were fixed using either BD fix (BD Biosciences) for 10 min at 4 °C or Foxp3/Transcription Factor Staining Buffer Set (ThermoFisher Scientific) for 60 min at 4 °C for intracellular staining. For intracellular staining, fixed and permeabilised cells were washed and stained with fluorochrome-conjugated antibodies against intracellular markers. Cell acquisition was performed using an LSRII (BD Biosciences) or LSRFortessa (BD Biosciences) flow cytometer. The data were analysed using FlowJo 10.5.3.

**Mass cytometry.** Some metal-conjugated antibodies were purchased from Fluidigm. Some purified unlabelled antibodies were conjugated in-house using the Maxpar X8 Metal Labelling Kit (Fluidigm) according to the manufacturer's instructions. All antibodies were titrated to determine the optimal staining concentration using murine splenocytes, aortic cells, and BM cells. The antibodies used are listed in Supplementary Table 7. Single cells from murine tissues were first stained with a Rhodium DNA intercalator (Fluidigm) to distinguish live or dead cells followed by barcoding using the palladium-based 20-Plex Pd Barcoding Kit (Fluidigm) according to the manufacturer's instructions. Barcoded cells were then combined into a single tube prior to addition of rat anti-mouse CD16/32 (BD Biosciences) and staining with metal-conjugated antibodies for 30 min at 4 °C. The cells were then washed and were permeabilised using the Foxp3/Transcription Factor Staining Buffer Set (ThermoFisher Scientific) for 60 min at 4 °C prior to staining with antibodies against intracellular markers for 30 min at 4 °C. Finally, the stained cells were incubated with an Iridium DNA intercalator (Fluidigm) in Maxpar fix and perm buffer (Fluidigm) overnight at 4 °C. Prior to acquisition, cells were washed twice with cell staining buffer (Fluidigm) followed by two washes with water (Fluidigm) and filtered through 40 μm cell strainer (Corning) before introducing into a Helios mass cytometer (Fluidigm). Data were generated in an fcs file format. All fcs files in the experiment were normalised and debarcoded using the normalisation and debarcoding tools within the Helios software. These fcs files were then uploaded to Cytobank (www.cytobank.org) for all gating and further analysis using the automated dimensionality-reduction algorithm tSNE[12].

**Cell culture.** BM cells were cultured in RPMI-1640 (Gibco) with 10 % FBS (Bio-Sera), 10 units/ml penicillin (Gibco), 0.1 mg/ml streptomycin (Gibco), 2.5 μg/ml amphotericin B (Gibco) and 50 μM 2-mercaptoethanol (Gibco), together with 20 ng/ml CSF2 (Peprotech) or 100 ng/ml CSF1 (Peprotech). The medium was refreshed every 3 to 4 days. CSF2-cultured macrophages were gated as CD11b+CD11c+F4/80+ MHCIIlow-int as published[60]. For macrophage activation, adherent CSF1-BMDMs were replated with the medium containing appropriate stimuli for 18 h. Ligand concentrations used for macrophage priming are 100 ng/ml LPS (Enzo Life Sciences), 100 ng/ml IFNγ (Biolegend), 20 ng/ml IL-4 (Peprotech), 20 ng/ml IL-13 (Peprotech), and 100 ng/ml FSL-1 (Invivogen). For SHP1 inhibition assays, 100 ng/ml TPI-1 was treated for 2 days for foam cell assay. For proliferation assays, cells were treated with

10 μM EdU for 4 h and were stained using EdU Staining Proliferation Kit (Abcam) according to the manufacturer's instructions.

**Tissue and cell coculture**. Heart, lungs and aorta were dissected and were rinsed in PBS (Lonza) to remove the excessive blood. Finely cut tissues were placed in RPMI containing 1% FBS (Bio-Sera), 10 units/ml penicillin (Gibco), and 0.1 mg/ml streptomycin (Gibco), and were then transferred to an insert containing 0.5 μm-sized pores (Greiner Bio-One). This insert allows migration of small molecules such as cytokines and growth factors while blocking cell migration. The inserts were placed in the 6 well plate where cells were seeded a day before, and tissues and cells were cocultured at 37 °C for 24 h. The insert containing the tissues was removed and the cells were washed in PBS. Cells were lysed in 700 μl of RLT buffer (Qiagen) for gene expression assay.

**Foam-cell assay**. CSF1-cultured BMDMs were plated on coverslips in 10% lipoprotein-deficient serum (Sigma-Aldrich) in RPMI overnight before being treated with 25 μg/ml oxidised low-density lipoprotein (oxLDL) (ThermoFisher Scientific) for 24 h. Cells were washed with PBS twice and stained with 5 μM BODIPY™ 493/503 (ThermoFisher Scientific) in PBS for 30 min at 37 °C. After two wash steps, cells were fixed in 4% paraformaldehyde (PFA) (ThermoFisher Scientific) for 10 min at room temperature and were mounted with Fluoromount-G with DAPI (ThermoFisher Scientific).

**Cholesterol-efflux assay**. CSF1-cultured BMDMs were treated with 25 μg/ml oxLDL (ThermoFisher Scientific) for 24 h prior to cholesterol efflux assay. All procedures were performed using Cholesterol Efflux Assay Kit (Cell-based) (Abcam), according to the manufacturer's instructions.

**Enzyme-linked immunosorbent assay (ELISA)**. IL-6 in the supernatant from macrophage culture was measured using Mouse IL-6 DuoSet ELISA (R&D systems), according to the manufacturer's instructions.

**Gene expression assay**. Total RNA was extracted from cell lysates in RLT buffer using a RNeasy Mini kit (Qiagen) according to the manufacturer's instructions. About 50 ng/ml of RNA was synthesised into cDNA using a High Capacity cDNA Reverse Transcription kit (ThermoFisher Scientific). A pre-amplification step was performed according to the manufacturer's instructions using appropriate TaqMan assays (ThermoFisher Scientific) and TaqMan Pre-Amp Master Mix (Thermo-Fisher Scientific). The pre-amplified cDNA, TaqMan Assays, and TaqMan Gene Expression Master Mix (ThermoFisher Scientific) were used for qPCR in a 384-well plate and the reaction was run by Viia7 Real Time PCR system (ThermoFisher Scientific), according to the manufacturer's instructions. TaqMan probes used are *18 S rRNA* (Hs99999901_s1), *egfp* (Mr04097229_mr), *Hbegf* (Hs00181813_m1), *Clec4a2* (Mm04210363_mH), *Mrc1* (Mm00485148_m1), *Actb* (Mm02619580_g1), *Abca1* (Mm00442646_m1), and *Abcg1* (Mm00437390_m1).

**RNA sequencing**. Single-cell RNA sequencing was performed on live CD45+CD11b+ aortic cells from 9 *ApoE*−/− and *ApoE*−/− *Clec4a2*−/− female mice fed a HFD for 12 weeks. Briefly, cells were sorted using FACSAria III (BD Biosciences), washed in PBS with 0.04% BSA and resuspended before capturing single cells in droplets on a Chromium 10x Genomics platform. Library generation for 10x Genomics v2 chemistry was performed following the Chromium Single Cell 3′ Reagents Kits (User Guide: CG00052). Quantification of cDNA and library construction was performed using a Qubit dsDNA HS Assay Kit (ThermoFisher Scientific) and high-sensitivity DNA tapestation (Agilent). Libraries were sequenced on an Illumina HiSeq4000 platform to achieve an average of 50,000 or more reads per cell. The raw sequence data were generated in a FASTQ file format and the alignment data were generated in a BAM file format.

Poly(A) RNA sequencing was carried out on BM cells obtained from CD45.1 and *Clec4a2*−/− mice and cultured with 100 ng/ml CSF1 for 10 days. Macrophages were collected and sorted into CD11b+F4/80+CD206+ populations that then were separated by the expression of CD45.1 (WT) or CD45.2 (*Clec4a2*−/−). Total RNA was isolated by using the RNeasy micro kit (Qiagen). Total RNA quantity and integrity were assessed using Quant-IT RiboGreen RNA Assay Kit (ThermoFisher Scientific) and high-sensitivity DNA tapestation (Agilent). Purification of mRNA, generation of double-stranded cDNA, and library construction were performed using NEBNext Poly(A) mRNA Magnetic Isolation Module (New England Biolabs) and NEBNext Ultra II Directional RNA Library Prep Kit (New England Biolabs) for Illumina with adaptors and barcode tags (dual indexing)[61]. The final size distribution of the pool was determined using high-sensitivity DNA tapestation (Agilent), and quantification was determined by Quant-IT RiboGreen RNA Assay Kit (ThermoFisher Scientific) before sequencing on an Illumina HiSeq 4000 as 75 bp paired end.

Raw gene counts across multiple biological samples were pooled using the CellRanger command line interface. Reads were subsampled from higher-depth libraries until they all had an equal number of confidently mapped reads per cell. Integrated raw gene matrix was passed to Seurat R package (v 3.0)[62] for preprocessing and quality control. Quality control pipeline removed $n = 1,467,497$

cells that have fewer than 200 expressed genes and $n = 14,872$ genes that were detected in fewer than 3 cells. Cells with greater than 5% mitochondrial gene expression were also excluded from the analysis. The count matrix was log-normalised and scaled to unit variance, while mitochondrial and cell count effects were removed using a linear regression. Normalised count matrix was reduced to 50 principal components (PCs) and significant PCs [Jackstraw test[62]] were used as the input for graph-based clustering (as implemented in Seurat). Data visualisation and dimensionality reduction was achieved using three algorithms including tSNE[63], LargeVis[64], and ivis[65].

**Gene expression in human atherosclerosis**. Human-plaque RNA expression data were obtained from the Tampere Vascular Study[66]. The study was approved by the Ethics Committee of Pirkanmaa Hospital District (Finland) and informed consent was previously acquired from patients enroled in the study. Vascular samples were collected from patients subjected to carotid endarterectomy. Clinical characteristics of the study population are available in Supplementary Table 8.

Total RNA was extracted using a RNeasy Mini kit (Qiagen) according to the manufacturer's instructions. About 200 ng of RNA was synthesised into cDNA using the Ambion's Illumina RNA Amplification kit according to the instructions (Ambion). Microarray assays were performed using Illumina HumanHT-12 version 3 Expression BeadChip (Illumina), according to the manufacturer's instructions (Illumina). The gene-association data from the carotid samples were analysed using a Spearman's rank-correlation coefficient test.

**Murine atherosclerotic lesion assays**. All materials were purchased from Sigma-Aldrich unless stated otherwise. Frozen sections of the aortic root (5 μm each) were stained in Oil Red O solution (3 parts 1% w/v in isopropanol and 2 parts 1% w/v dextrin in distilled water) and counterstained with haematoxylin for measuring lesion formation. The whole aortas were dissected and fixed in 4% PFA. The fixed tissue was cut open longitudinally from the iliac arteries to the aortic root and pinned flat. The aortas were stained in the Sudan IV solution (0.5% w/v in equal parts 50% ethanol and acetone). Sudan IV-stained plaque area was measured. Hematoxylin and eosin (H&E) staining was used to determine necrotic areas in the lesion within the aortic root. The areas within the lesion that were free from H&E stain and bigger than 3000 μm$^2$ were designated to be necrotic core.

**Avidin–biotin complex (ABC) staining**. Immunohistochemistry of tissue sections was performed using the Avidin–Biotin Complex (ABC) kits (Vector Laboratories) based on the optimised protocol[67]. Acetone-fixed slides were washed in PBS and blocked with 10% serum (rabbit or goat serum from Abcam) for 1 h. After washing in PBS, Avidin and Biotin blocking solutions (Vector Laboratories) were used to block endogenous avidin and biotin for 15 min each. The slides were incubated with a primary antibody for 1 h and washed in PBS. The slides were then incubated with a biotinylated secondary antibody (Vector Laboratories) diluted by 1:400 in PBS for 1 h. After two washing steps, the slides were then treated with 0.3% hydrogen peroxide (Sigma-Aldrich) for 15 min to block endogenous peroxidase activity. The slides were then washed and incubated with an ABC-Elite-Standard Kit for Peroxidase (Vector Laboratories) for 40 min. After two washing steps, the slides were treated with DAB (3,3′-diaminobenzidine) HRP substrate solution (Vector Laboratories) for 1–5 min depending on the intensity of the signal. The slides were then washed in PBS, counter-stained in Haematoxylin solution (Sigma-Aldrich) for 15–30 s, and washed in deionised water. The slides were dehydrated in 70% ethanol for 1 min and 100% ethanol for 1 min. Finally, the slides were air-dried and were mounted using Histomount (National Diagnostics) and cover glasses (VWR).

**Immunofluorescence**. Acetone or PFA-fixed slides were washed in PBS and were blocked with 20% normal serum (rabbit or goat serum from Abcam) for 1 h. After washing steps in PBS, the slides were treated with primary antibodies for 1 h and were washed in PBS twice. The slides were then incubated with secondary antibody conjugated with AF488, AF568 or AF647 (10 μg/ml, ThermoFisher Scientific) for 1 h in the dark. The slides were then washed in PBS followed by dipping in deionised water briefly. The slides were then mounted with Fluoromount-G with DAPI (ThermoFisher Scientific).

**Imaging and analysis**. All stained sections were imaged using an Olympus BX51 osteometric bright-field and fluorescence microscope, and en face aorta images were taken using the Leica M60 dissection microscope. All images were randomised for blind analysis. Quantification of stained images were analysed using Clemex Vision Lite software. Some images were analysed using ImageJ 2.1.0/1.53c (NIH, USA).

**Serum cholesterol measure**. Levels of cholesterol in the murine blood were measured by using cholesterol-detection reagent (Randox) alongside the cholesterol standard (Randox). The enzymatic endpoint method was adopted to calculate the concentration of cholesterol based on the standard.

**Statistical analysis**. All statistical analyses were performed using GraphPad Prism. All data are presented as Mean ± SEM. Two tailed Student's *t*-test, two-way ANOVA or ordinary one-way ANOVA with Holm–Sidak's multiple-comparison test was performed for statistical significance. For the scRNA-seq analyses, differentially expressed genes were identified using the two-sided Wilcoxon Rank Sum test with Bonferroni's correction for multiple-hypothesis testing. In the bulk RNA-seq analyses, differential expression was assessed using the two-sided Robinson and Smyth exact test with Bonferroni's correction for multiple-hypothesis testing. $P < 0.05$ was considered statistically significant: $*P < 0.05$, $**P < 0.01$, $***P < 0.001$ and $****P < 0.0001$.

**Reporting summary**. Further information on research design is available in the Nature Research Reporting Summary linked to this article.

## Data availability

All data generated in this study are provided in the Supplementary Information files and the Source Data file. The scRNA-seq data of myeloid cells from murine atherosclerotic aortas are publicly available at ArrayExpress using the following accession codes (Fig. 1: E-MTAB-10743; Fig. 7: E-MTAB-10746). The bulk RNA-seq data of murine CSF1-cultured BM-derived macrophages in Fig. 5 are also freely accessible (accession code: E-MTAB-10734). The data included in Supplementary Fig. 4 are obtained from the Tampere Vascular Study (TVS)[66], which comprises health related participant data. The use of data is restricted by regulations on professional secrecy (Act on the Openness of Government Activities, 612/1999) and sensitive personal data (Personal Data Act, 523/1999, implemented in the EU data-protection directive 95/46/EC). Due to these restrictions, the data cannot be stored in public repositories or otherwise made publicly available. Data access may be permitted on a case-by-case basis upon request. Data sharing outside the group is done in collaboration with the TVS group and requires a data-sharing agreement. Investigators can submit an expression of interest to the Chairman of the publication committee (Prof Terho Lehtimäki, Department of Clinical Chemistry, Fimlab Laboratories and Finnish Cardiovascular Research Center-Tampere, Faculty of Medicine and Health Technology, Tampere University, Arvo building room D338, P.O Box 100, FI-33014 Tampere, Finland; email: terho.lehtimaki@tuni.fi; phone: +358 50 4336285). The timeframe for response to request is within two weeks. The right to use the data is granted with the understanding that the collaborator will protect the data and not share it with any other parties, academic or commercial, other than those mentioned in the agreement. The data can be used by the collaborator only for scientific purposes and as specified in the appendix of the agreement in the form of a short scientific proposal. After the contract has ended, the collaborator will upon directions of TVS, return or destroy copies of the data. In case the work has involved transfer of bio-specimens, the collaborator will return the remaining biospecimens and related data to the TVS group. Source data are provided with this paper.

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

## Acknowledgements

We would like to thank Dr David Ahern (University of Oxford) for acquisition and guidance in analysing the mass cytometry data, Associate Professor Tal Arnon (University of Oxford) for guidance on the DT ablation and data interpretation, Dr Emma Raitoharju (University of Tampere) for the human transcriptomic data, Professor Niku Oksala (University of Tampere) for collection of human tissues; Dr Leena Viiri (University of Tampere) for a pilot study on CLEC4A2, Professor Yoichiro Iwakura (University of Tokyo) for providing *Clec4a2*$^{-/-}$ mice and Professor Naoki Matsumoto (University of Tokyo) for providing CLEC4A2 antibody, Mr Jonathan Weber for the FACS sorting service, and The Oxford Genomics Centre for the RNA sequencing service. We are grateful to Professor Fiona Powrie (University of Oxford) and Professor Siamon Gordon (University of Oxford) for scientific discussions and Dr Elizabeth Thompson (University of Oxford) for helpful suggestions. C.M. was supported by the European Commission under the Seventh Framework Programme (grant agreement n° 201668 [Athero-Remo], FP7/2007–2013; grant agreement n°HEALTH-F2-2013-602114 [Athero-B-Cell] and grant agreement n° HEALTH-F2-2013-602222 [Athero-Flux]) and the Novo Nordisk Foundation (Grant number NNF15CC0018346; NNF0064142). I.P. received a DPhil studentship from the Kennedy Trust for Rheumatology Research. Tampere Vascular study was supported with grants from the Competitive Research Funding of the Tampere University Hospital (Grant 9M048 and 9N035 for T.L.), the Emil Aaltonen Foundation (T.L.), the Pirkanmaa Regional Fund of the Finnish Cultural Foundation, the Research Foundation of Orion Corporation, the Jenny and Antti Wihuri Foundation, the Finnish Foundation for Cardiovascular Research, the Yrjö Jahnsson Foundation, European Union 7th Framework Program (grant 201668 for AtheroRemo), EU Horizon 2020 (grant 755320 for TAXI-NOMISIS), and the Academy of Finland grant 322098.

## Author contributions

Design of the project: I.P. and C.M. Data acquisition: I.P., M.G., J.C., N.Z. and L.P.L. Data interpretation: I.P. and C.M. RNA sequencing analysis: I.D. Human data analysis: L.P.L. and T.L. Design of mouse strain: C.M., J.C., E.A. and I.U. Preparation of the paper: I.P. and C.M. Review of the paper: I.P., C.M., J.C., M.G., N.Z., L.P.L., T.L., E.A., M.F., I.U. and I.D.

## Competing interests

The authors declare no competing interests.
