## [Peer Review File · Nature Communications]

C-type lectin receptor CLEC4A2 promotes tissue adaptation of macrophages and protects against atherosclerosisREVIEWER COMMENTS

Reviewer #1 (Remarks to the Author):

This is the first paper to report on the expression and role of Clec4a2 in “resident” macrophages and its implications for the development of atherosclerosis. The data suggest that Clec4a2 expression in macrophages regulates macrophage phenotype towards a resident-like anti-inflammatory phenotype with enhanced cholesterol efflux capacity. This is protective against atherosclerosis.

There are however important shortcomings. The genetic (DTR-mediated) depletion experiments may have induced important side effects unrelated to Clec4a2, and the genetic deletion experiments using whole KO of Clec4a2 were not specific to macrophages (whether resident or not). Therefore, the increase of lesion size in those experiments may be unrelated to Clec4a2 and its function in resident macrophages. It is also difficult to disentangle the impact of lesion size per se (versus Clec4a2 deficiency itself) on the distribution and identity of the different immune cell clusters in the aorta. Mass cytometry and single cell experiments shown in Figures 4 and 7 should have been done (and would have been more informative) on lesions with similar sizes collected from the 2 groups of mice.

Specific comments:

Suppl fig 1: CD206+CD169+ isn't enough to characterise what others call resident vascular macrophages. CD206 can be re-expressed on monocyte-derived macrophages without concomitant expression of other resident macrophage markers.

Figure 1: Clusters in which CLEC4A2 is expressed are C1, C3, C5 (and C11?). There's no GO analysis of C3 (should be provided), which seems to be also quite different from C1 and C5. The latter 2 clusters share a lot of similarities and it doesn't seem that C5 is very distinct from C1. I appreciate that Clec4a2 expression distinguished the 2 (C1 and C5) out of 3 “resident” clusters (C0, C1 and C5). However, Clec4a2 was also highly expressed in C3, which isn't a resident-like macrophage cluster (therefore, Clec4a2 isn't specific to the resident type of macrophages), and it's unclear whether Clec4a2 expression in C3 cluster was also inversely associated with lower expression of inflammatory genes. C3 appears to express high levels of Ly6c2 and Ccr2 and appears more like classical monocytes. It also appears that Clec4a2 is expressed in C11, which co-express IFN genes?

Suppl Fig 1b: how were the 170 inflammatory genes identified? And did you consider all inflammatory genes of each cluster? Is there a (inverse) relationship between inflammatory gene expression and Clec4a2 expression in each individual cluster?

Figure 2: Depletion of aortic CLEC4A2+ macrophages is very transient. They are already reconstituted by day 5

There's transient replenishment by Ly6Chi monocytes. But do those monocytes replenish the same niche, which is in the adventitia, not the intima?

Are those same monocytes go on to become CLEC4A2+ macrophages within 5 days? Or is there some proliferation of resident CLEC4A2+ macrophages that had escaped depletion?

If the newly recruited monocytes become CLEC4A2+ adventitial macrophages, is this also true in a setting of hypercholesterolemia (Apoe^{-/-} or Ldlr^{-/-} on HFD)?

Neutrophils are being depleted in BM and blood for at least 30 hours. Their nb in blood is still substantially lower compared to saline (and also compared to DT-treated LysM-WT/Clec4a2^{flox}/DTR mice?) at one week of DT treatment. No information is provided beyond this first week. Neutrophils may well be depleted continuously, not only in blood, but also in BM and spleen, with continuous DT treatment.

Does the killing of macrophages or neutrophils induce “non-specific” inflammation (unrelated to or super-imposed on to the atherosclerosis process), unrelated to CLEC4A2 function or to the functions of those cells? (Necrotic) cell death-induced inflammation? Is there any sign of systemic inflammation due to the killing of these cells?

Figure 2: Loss of resident macrophages should lead to vessel dilatation, according to previous work. Have you seen this?

The collar model is not analysed in detail. It's unclear what contributed to the increase of neointima formation.

BM transplant atherosclerosis experiment: irradiation will have depleted resident macrophages. Do those resident macrophages reconstitute (in the adventitia) from circulating monocytes before the start of HFD?

Again, it's important to see data on what chronic killing of a huge number of cells throughout the body is doing to the general inflammatory response.

Plasma lipid levels are not provided. Atherosclerosis was assessed only in en-face aorta.

Figure 4: Here the experiments involve whole KO of Clec4a2. Therefore, they are not specific for macrophages. A better experiment would be LysM-cre/Clec4a2floxed mice crossed to apoe^{-/-}. The increase of lesion size in Clec4a2 KO cannot be attributed to changes in macrophage subsets or function. The difference in aortic immune cells subsets between the 2 groups of mice could just reflect the difference in lesion size at the end of the experiment. Indeed, progression of atherosclerosis is known to be associated with reduced M2-like macrophages and increased M1-like macrophages.

Also, the use of Lyve1-cre/Clec4a2floxed mice could be more appropriate to address the role of Clec4a2 more specifically in resident-like macrophages.

Figure 5: It would be informative to get these BMDM stimulated using oxLDL or other atherosclerosis-relevant stimuli and assess how Clec4a2 deficiency would impact on their response/phenotype.

Figure 6: these experiments indicate that the deletion of Clec4a2 in all BM-derived cells could impact not only macrophages but also DC numbers and functions. Thus, the experiment in Figure 4 doesn't allow for any valid conclusion to be drawn regarding the contribution of macrophage-Clec4a2 to the observed atherosclerotic phenotype.

Experiments in Figure 4 and 6 are better done using LysM-cre/Clec4a2floxed mice.

Figure 7: The finding that deletion of Clec4a2 affects only one subset out of 3 of 4 subsets expressing Clec4a2 is not highly supportive of a critical role of Clec4a2 in the establishment of a resident-like macrophage phenotype. I'm wondering whether you would see the same difference in immune cell distribution just by comparing plaques of different size (at different stage of development).

One way to show that Clec4a2 deficiency directly affects the subsets of macrophages within the lesions is to repeat the experiment at an earlier time point at which no difference in lesion size can be detected between the 2 groups.

Would the finding that Clec4a2 deficiency promotes both foam cell formation and a pro-inflammatory phenotype be counter-intuitive knowing that most foam cell macrophages in atherosclerosis display an anti-inflammatory phenotype?

Reviewer #2 (Remarks to the Author):

This manuscript by Monaco and colleagues identifies the C-type lectin CLEC4A2 as a defining marker of vascular resident macrophages and demonstrates its essential role in deciding the fate of monocytes as they differentiate in situ. The role of CLEC4a2 in both vascular homeostasis and disease (arterial injury and atherosclerosis) is investigated using macrophage-specific deletion (LysMCreClec4a2flox-DTR) and whole body knock-out mouse models through a combination of mass cytometry, scRNA-seq and conventional histological analyses. The work is truly a tour de force and the manuscript is clearly written. The authors should be commended for their clear distillation of complicated datasets, and the integration of multiple independent techniques to demonstrate the role of CLEC4a2. I have only minor suggestions to improve the clarity of the results.

1. The authors begin by defining the macrophage populations in the intima-media and adventitial layers of aortas of ApoE^{-/-} mice, and demonstrate that the majority of CD206+CD169+ macrophages reside in the adventitia, while CD11c+CD44+ macrophages predominantly reside in the intima. In subsequent studies, this division is no longer addressed and the reader is left to guess that all further studies use the whole aorta. This should be clarified.
2. In the results section describing LysMCre-Clec4a2^{flox}-DTR mice, the expression of eGFP in various immune cell subtypes is described without any previous mention of how eGFP is incorporated into the LysMCre-Clec4a2^{flox}-DTR mice or what it is reporting. This should be clarified.
3. On page 10, the sentence that begins "The most notable change caused by CLEC4A2..." needs to be rewritten.
4. In the discussion, the authors provide a potential mechanism through which CLEC4a2 may regulate CSF2-mediated fate decisions, via its interaction with SHP1. Studies of SHP1^{-/-} macrophages showing that they phenocopy Clec4a2^{-/-} macrophages would have strengthened this argument. Given the large amount of data already in the manuscript, these experiments are not required, but would have helped close the "mechanistic loop".

Reviewer #3 (Remarks to the Author):

Previously the authors discovered two major macrophage subsets in the atherosclerotic aorta of mice: CD206+CD169+ and CD11c+CD44+. In this follow up study, the authors find that CD206+CD169+ abundance is lower in atherosclerotic aortas compared to WT and that Clec4a2 critically modulates this process. Using single cell RNA seq, their initial clue into this process is that Clec4a2 expression in macrophages from atherosclerotic aortas is negatively correlated with an inflammatory gene signature. Using a Cre-floxed DTR system to selectively ablate Clec4a2 macrophages and partial BM chimera to KO Clec4a2, the authors found that atherosclerotic lesions are significantly larger and more necrotic in the absence of Clec4a2. Further, in the absence of Clec4a2, macrophages were: i) more likely to differentiate into Ly6c+ macrophages, ii) sensitized to TLR stimulation; iii) lacking lipid metabolism genes; iv) poorly equipped for cholesterol efflux. The authors deduce that Clec4a2 promotes monocyte differentiation into resident macrophages in healthy and atherosclerotic aortas and thus protects the aorta from atherosclerotic lesions.

Noteworthy results include:

1. Clec4a2 expression is inversely associated with an inflammatory signature in scRNA-seq.
2. Atherosclerotic lesions are bigger and more necrotic in the absence of Clec4a2+ macrophages,
3. Clec4a2 expression biases differentiation: Clec4a2 +/+ (WT) more likely to differentiate into res macs; Clec4a2 -/- more likely to differentiate into CD11c, Ly6c, cDC2.
4. Macrophages missing Clec4a2 downregulate lipid metabolism genes and thus may poorly process lipids or export cholesterol, possibly contributing to larger lesions.
5. Macrophages missing Clec4a2 are sensitized and express more IL6 in response to TLR ligands

The work in this paper is of very high quality and supports the majority of the conclusions. There is enough supporting information in the methods for reproduction (except I could not find the 170-gene inflammatory signature)

As for significance, their finding that Clec4a2 expression is negatively correlated with an inflammatory signature is critical, and led them to delete Clec4a2 in macrophages and discover an important but previously unknown role for Clec4a2 in tissue macrophage homeostasis. Their results are significant for the field of atherosclerosis and macrophage biology and could also impact numerous areas of biology if Clec4a2 has a similar role in other damaged organs and if the results extend to human disease.

Criticisms:

After culturing CSF1- and CSF2-BMDM from WT and KO mice ex vivo, it wasn't clear what they differentiate into. Do the differentiated states match those identified in vivo from KO vs WT BM transfers? Are the "core" gene sets identified in vivo similar in the cultured cells? As the figure and text stands, one interpretation of these data is that CSF1 induces WT cells to proliferate, and that

CSF2 induces Clec4a2 ^{-/-} cells to proliferate. But the text does not make it clear whether these cells differentiate into a relevant state or express (or lack) gene programs identified in vivo. Thus, while this is an interesting observation ex vivo, the results do not link up yet with the results in vivo. Still, the finding that CSF2-induced TFs are upregulated in vivo (as mentioned in discussion) does provide some more support for the theory. Future experiments with CSF2 deficient mice, or other manipulations of CSF1 and 2, may help clarify these questions.

POINT-BY-POINT RESPONSE TO THE REVIEWERS

We would like to thank the Reviewers for thoroughly reviewing our manuscript, and for their positive comments and insightful critique on various specific matters. Our revised manuscript is now very much improved, addressing all the issues raised. It includes a significant amount of new data and analyses, together with amendments of the text. A detailed point-by-point reply to the Reviewers' comments is provided below.

Reviewer #1 (Remarks to the Author):

This is the first paper to report on the expression and role of Clec4a2 in “resident” macrophages and its implications for the development of atherosclerosis. The data suggest that Clec4a2 expression in macrophages regulates macrophage phenotype towards a resident-like anti-inflammatory phenotype with enhanced cholesterol efflux capacity. This is protective against atherosclerosis.

Thank you for the Reviewer for highlighting the novelty of our manuscript. We have respectfully added numbering to the review to facilitate cross-referencing.

1) There are however important shortcomings. The genetic (DTR-mediated) depletion experiments may have induced important side effects unrelated to Clec4a2,...

We agree with the Reviewer that this is a point of major importance. We have, however, carefully controlled for these side effects of DT-mediated ablation in our experiments in several ways. First, we have used a concentration of DT that is commonly used in the literature with no gross effects usually observed. Moreover, we have generated and employed a DTR model where DTR is specifically induced in cells expressing both *LysM* and *Clec4a2* and therefore only a limited population of cells is targeted; this should limit side effects as it is well known that the larger the population in the body DT depletes, the larger the side effects. We have included further experiments in the revision as suggested by the Reviewer to strengthen our evidence of specificity of ablation in the blood, spleen and bone marrow (discussed at point 11).

If apoptosis-induced inflammation indirectly contributes to an increase in lesion size, we would expect that DT based depletion of other myeloid cells in this context will lead to similar results. However, this is not the case. For example, previous landmark papers used the CD11b^{DTR} model to deplete major myeloid subsets in atherosclerosis (DOI:10.1161/01.RES.0000260802.75766.00). Under these conditions, the opposite effect was reported at a comparable time-point to the one we used in our study: a decrease in lesion size. Given that CD11b⁺ cells are much more abundant in the aortic lesion and throughout the body than *LysM Clec4a2* expressing cells, it is expected that if non-specific apoptotic-mediated effect induced by DT treatment would have contributed to an increase in lesion size, such models would have revealed it.

Second, we have measured sensitive markers of inflammation such as IL-6 in the serum of *LysM^{Cre+} Clec4a2^{flox/DTR}* mice following treatment with DT, as a measure of inflammation. No increase in IL-6 levels was detectable compared to the control baseline (see below the graph and also added in Supplementary Table 4), reflecting the fact that a relatively low number of cells are being ablated due to the refined genetic approach we employed. These data are discussed in the Results (Page 10, Line 25) and in the Discussion (Page 18, Line 4).

Third, our data do not rely only on the DT-depletion experiments but rather a combination of *in vivo* and *in vitro* systems: 1) atherogenesis study in the *ApoE^{-/-}* and *ApoE^{-/-} Clec4a2^{-/-}* mouse strains that leads to a significant loss of resident vascular macrophages (see Point

2), 2) competitive chimera studies in the steady state and atherosclerotic conditions, showing a consistent effect on resident vascular macrophages when cell extrinsic factors are controlled for, 3) extensive scRNA-seq and bulk RNA-seq data that detail the specific effects of this pathway *in vitro* and *in vivo*. All evidence from complementary experimental models and technical approaches consistently converges on an athero-protective role of CLEC4A2 in arterial macrophages and the role of CLEC4A2 in supporting their representation and function during atherogenesis.

Serum IL-6 levels in *Ldlr*^{-/-} chimera mice reconstituted with the bone marrow of *LysM*^{Cre+} *Clec4a2*^{flox/DTR} or *Clec4a2*^{flox/DTR} mice. The mice were fed an HFD and treated with saline or DT for 12 weeks. Data are presented as mean±SEM. Ordinary one-way ANOVA with Holm-Sidak's multiple comparisons test. Non-significant. (Data are now added in Supplementary Table 4).

2)... and the genetic deletion experiments using whole KO of *Clec4a2* were not specific to macrophages (whether resident or not). Therefore, the increase of lesion size in those experiments may be unrelated to *Clec4a2* and its function in resident macrophages. It is also difficult to disentangle the impact of lesion size per se (versus *Clec4a2* deficiency itself) on the distribution and identity of the different immune cell clusters in the aorta. Mass cytometry and single cell experiments shown in Figures 4 and 7 should have been done (and would have been more informative) on lesions with similar sizes collected from the 2 groups of mice.

We acknowledge the general limitation of using a global KO. Unfortunately, a *Clec4a2*^{flox-KO} mouse strain does not exist. To generate it and perform all the crosses with *LysM*^{Cre} and the *ApoE*^{-/-} background would take years and would not have been available in time for this revision. For this reason, we have used a combination of models and techniques to obtain this information.

First, the CLEC4A2-DT depletion model provides information about the protective effect of CLEC4A2⁺ macrophages on the development of atherosclerosis (Figure 3, point 1 above). We have included further experiments in the revision as suggested by the Reviewer to strengthen our evidence of specificity of ablation in the blood, spleen and bone marrow (discussed at point 11).

Second, CLEC4A2 genetic deletion alters the representation of CLEC4A2⁺ vascular macrophages during atherogenesis without changes to circulating populations in the peripheral tissues. We simultaneously monitored cells from the BM, blood, spleen and para-aortic lymph nodes using mass cytometry, and the data showed no increase in any immune cell subsets in these tissues (Supplementary Table 7). This evidence indicates that the CLEC4A2 protects from atherogenesis through its effects within the arterial wall.

Third, we performed a new atherogenesis study with chow-diet fed *ApoE*^{-/-} and *ApoE*^{-/-} *Clec4a2*^{-/-} mice to investigate earlier stages of atherosclerosis, suggested by the Reviewer (added in Supplementary Figure 7 and pasted below this response for the Reviewer's convenience). We found that the lesion size was not statistically different between the two groups. Mass cytometry analysis of the aortic immune cells specifically revealed a profound reduction in resident Lyve1⁺ vascular macrophages in CLEC4A2-deficient *ApoE*^{-/-} mice. No differences were observed in other myeloid cells including CD11c⁺ macrophages, monocytes and neutrophils, cDC1 and cDC2 or lymphocytes in *ApoE*^{-/-} *Clec4a2*^{-/-} mice. These findings are consistent with the evidence that the ablation of the CLEC4A2⁺ macrophages in the experiments with the *LysM*^{Cre+} *Clec4a2*^{flox/DTR} mice has a pro-

atherogenic effect (Figure 3). These new data suggested by the Reviewer strengthen the main conclusions of our work by showing that the primary function of CLEC4A2 is to preserve the size of the athero-protective resident vascular macrophage pool during atherogenesis, and that this loss precedes the changes in lesion formation and local myeloid cell profile. A concomitant dilatation of the vessel wall in *ApoE^{-/-} Clec4a2^{-/-}* mice was a sign of loss of resident macrophages in previous studies (DOI:10.1016/j.immuni.2018.06.008, as noted by the Reviewer in point 13), also further supports this conclusion.

Finally, these data are substantiated by other observations in competitive chimera experiments (Figure 6), demonstrating that the effects of CLEC4A2 are cell-intrinsic and ascribable to a single overarching mechanism of a bias from CSF2 to CSF1-directed monocytic fate decision in the aorta. These findings match the transcriptional signature found *in vivo* in the aorta (Figure 7), demonstrating that CLEC4A2 plays a major role in protecting the representation and phenotype of vascular resident macrophages in atherosclerosis.

Together, these lines of evidence demonstrate that CLEC4A2⁺ macrophages are athero-protective (Figure 3), and that CLEC4A2 is essential for their representation in aortic tissues during atherogenesis, irrespective of cell extrinsic factors (competitive chimera in Figure 6) and lesion size (Supplementary Figure 7). In summary, our study establishes an important role for CLEC4A2 in avoiding the excessive loss of athero-protective vascular macrophages during atherogenesis. We have also clarified these points in the Discussion (Page 19 Line 17).

Supplementary Figure 7 (related to Figure 4).

a, Schematic showing experimental set up to compare 20-week old $ApoE^{-/-}$ and $ApoE^{-/-} Clec4a2^{-/-}$ littermate mice fed a chow diet for 12 weeks. **b**, Representative images and plaque quantification of Oil Red O-stained aortic roots of $ApoE^{-/-}$ and $ApoE^{-/-} Clec4a2^{-/-}$ littermate mice ($n=13$ each; Scale bar: $200\mu m$). **c**, Representative images of aortic roots showing arterial size (dotted area) (Scale bar: $200\mu m$). **d**, Representative images of aortic root lesions stained with haematoxylin and eosin (H&E) and the size of necrotic core (H&E-free area; dotted) (Scale bar: $100\mu m$). **e**, Representative images and quantification of CD68-stained (brown)-area in aortic root lesions (dotted). (Scale bar: $100\mu m$). **f**, tSNE clustering of live $CD45^{+}$ cells from whole aortas of chow diet fed $ApoE^{-/-}$ and $ApoE^{-/-} Clec4a2^{-/-}$ mice by mass cytometry. Leucocyte populations were identified based on expression of 32 markers. **g, h**, Frequency of aortic leucocyte populations in each genotype ($n=6$; pooled from two independent experiments). All data are presented as mean \pm SEM. Two-Way ANOVA (b) or Two tailed Student's t test. $**P < 0.01$.

Specific comments:

3) Suppl fig 1: $CD206+CD169+$ isn't enough to characterise what others call resident vascular macrophages. $CD206$ can be re-expressed on monocyte-derived

macrophages without concomitant expression of other resident macrophage markers.

We apologise for the confusion caused by this labelling and we agree that CD206 and CD169 expression alone does not fully define resident macrophages. Unbiased clustering via tSNE analysis of the mass cytometry data in Supplementary Figure 1 does not rely on individual marker expression but on a combination of markers to uncover clusters. We have added the histology and mass cytometry data of Lyve1 expression alongside CD206 and CD169 in Supplementary Figure 1g,i to show that the adventitial macrophage clusters express Lyve1, CD206 and CD169.

Our paper does not solely rely on mass cytometry, but we used scRNA-seq to better understand the macrophage transcriptional states. Our scRNA-seq data in Figure 1 and Supplementary Table 1 show that Lyve1 is expressed by three resident macrophage clusters, together with CD206. We added this clarification to the Results section (Page 5 Line 6).

4) Figure 1: Clusters in which CLEC4A2 is expressed are C1, C3, C5 (and C11?). There's no GO analysis of C3 (should be provided), which seems to be also quite different from C1 and C5.

We apologise for the omission. We have now included the GO pathways of all clusters in Figure 1 and Supplementary Figure 2c (see below). The Reviewer is absolutely right in saying that C3 represents Ly6C⁺ monocytes since it expresses monocyte markers including *Ly6c2* and *Ccr2*. We have now added additional cluster notations in the Results (Page 5 Line 14).

5) ...The latter 2 clusters share a lot of similarities and it doesn't seem that C5 is very distinct from C1.

We previously identified two resident-like macrophages in the healthy and atherosclerotic artery by using mass cytometry: CD209b⁺ and CD209b⁻ subsets (DOI:10.1093/cvr/cvy109). Unbiased clustering of our scRNA-seq data in Figure 1 also detected these two subpopulations. Gene ontology analysis of the 2 subsets suggests that potential cellular functions may be to some extent distinct (Figure 1e). Since clustering algorithms both in mass cytometry and scRNA-seq distinguish the two clusters, we report them as separate subsets.

6) I appreciate that Clec4a2 expression distinguished the 2 (C1 and C5) out of 3 "resident" clusters (C0, C1 and C5). However, Clec4a2 was also highly expressed in C3, which isn't a resident-like macrophage cluster (therefore, Clec4a2 isn't specific to the resident type of macrophages), and it's unclear whether Clec4a2 expression in C3 cluster was also inversely associated with lower expression of inflammatory genes. C3 appears to express high levels of Ly6c2 and Ccr2 and appears more like classical monocytes.

As shown in Supplementary Table 1, *Clec4a2* is expressed by Lyve1⁺ macrophage clusters C1 and C5 with statistically significant difference compared to all the other clusters in the analysis. *Clec4a2* expression is low and not statistically significant in C3. Low expression at the protein level is also evident in Supplementary Figure 3b,c.

We performed additional correlation analysis between inflammatory genes and CLEC4A2 expression. We found strong negative associations in C1 and C5 and a weaker correlation in C3. These data are now added to Supplementary Figure 2c. Please see also Point 8, where the same comment is also discussed.

7) It also appears that *Clec4a2* is expressed in C11, which co-express IFN genes?

A small (and non-significant; Figure 1) expression of *Clec4a2* is also seen in interferon (IFN) responsive macrophages. However, this small population of arterial macrophages (<1% of total CD11b⁺ myeloid cells) is unaffected by CLEC4A2 deficiency in the atherosclerotic mice (Figure 7a,b), suggesting that CLEC4A2 has a limited role in their functions.

8) Suppl Fig 1b: how were the 170 inflammatory genes identified? And did you consider all inflammatory genes of each cluster? Is there a (inverse) relationship between inflammatory gene expression and *Clec4a2* expression in each individual cluster?

We agree that adding the full list of inflammatory genes analysed will be helpful and informative and have now added it in Supplementary Table 2. The list includes selection of genes relevant to inflammation and macrophage biology that were identified in our RNA-seq datasets (reported in Supplemental Table 1, 8 and 9).

We very much appreciate the reviewer's suggestion to test if there is an inverse association between CLEC4A2 and inflammatory genes in each cluster expressing *Clec4a2*. We performed the analysis and found strong negative associations in C1 and C5 (Supplementary Figure 2c; see below). A weaker correlation is noted in C3, possibly due to its low expression of CLEC4A2. C11 was excluded from this analysis because its expression of *Clec4a2* was low. This observation adds support to our hypothesis that CLEC4A2 acts to inhibit inflammatory responses and is now discussed in the Results (Page 6 Line 19).

Supplementary Figure 2 (related to Figure 1).

Total cells from **whole aortas** of 9 *ApoE*^{-/-} mice fed an HFD for 12-16 weeks were sorted as live CD45⁺CD11b⁺CD64^{int-high} (first experiment) or live CD45⁺CD11b⁺ (second and third experiments) populations. These cells were analysed by single cell RNA-sequencing (scRNA-seq) using the 10X genomics platform. **a**, Total cell numbers and the proportion of each cluster in three independent experiments. **b**, The Gene Ontology (GO) enrichment analysis of clusters based on differentially expressed genes in myeloid clusters. Enriched pathways are presented as $-\log_{10}(\text{Fisher } P)$ using topGO analysis. **c**, Genetic association between *Clec4a2* and the average expression of 170 inflammatory genes in Cluster 1, 3 and 5.

9) Figure 2: Depletion of aortic CLEC4A2+ macrophages is very transient. They are already reconstituted by day 5.

There's transient replenishment by Ly6Chi monocytes. But do those monocytes replenish the same niche, which is in the adventitia, not the intima?

CD11c⁺ intimal macrophages are not depleted when DT is administered in the *LysM*^{Cre+} *Clec4a2*^{flox/DTR} mouse strain at the steady state nor in atherosclerotic aorta (Figure 2d, Figure

3b). We have now clarified this in the Results (Page 8 Line 25 and Page 10 Line 20). To answer the Reviewer's comment, we performed new experiments that are now included in Figure 2b-d and Supplementary Figure 6. The new experiments better capture the kinetics of depletion and replenishment of resident vascular macrophages by monocytes. We were able to show that resident macrophages are ablated from Day 1 to Day 4 (Figure 2b). They are reconstituted in the adventitia at Day 6 both in the steady state and atherogenesis (Supplementary Figure 6j,k; see also point 10 below). Thus, the niche that is affected by the depletion (and which is subsequently reconstituted) primarily overlaps with the adventitial niche, not the intimal niche.

This repopulation kinetics of depleted tissue macrophages by recruited monocytes is not uncommon and already detailed in other organs (DOI:10.1126/science.aau0964). A similar repopulation has been described in the artery at birth (DOI:10.1038/nr3343). The realisation that the depletion of CLEC4A2⁺ macrophages was transient is the primary reason that we adopted a chronic DT administration strategy to study the role of CLEC4A2⁺ macrophage on lesion formation (Figure 3). A chronic DT strategy is widely used in previous studies of atherogenesis and we modelled protocols from those studies (DOI:10.1161/01.RES.0000260802.75766.00; DOI:10.1161/CIRCRESAHA.109.210781).

10) Are those same monocytes go on to become CLEC4A2+ macrophages within 5 days?

Or is there some proliferation of resident CLEC4A2+ macrophages that had escaped depletion?

If the newly recruited monocytes become CLEC4A2+ adventitial macrophages, is this also true in a setting of hypercholesterolemia (ApoE^{-/-} or Ldlr^{-/-} on HFD)?

To first investigate whether the same monocytes become CLEC4A2⁺ macrophages post ablation, we performed adoptive transfer of Ly6C⁺ monocytes from *Cx3cr1^{eGFP+}* CD45.1⁺CD45.2⁺ mice to healthy and atherosclerotic mice (*LysM^{Cre} Clec4a2^{flox-DTR}* CD45.2⁺ or *Ldlr^{-/-}* CD45.1 chimera) post macrophage depletion. We find that transferred monocytes are readily detected in the peripheral blood and we were able to trace expression of GFP and CD45.1 in reconstituted macrophages in healthy and atherosclerotic aortas by flow cytometry (Supplementary Figure 6a-g). In addition, we also detected cells co-expressing CD64 and GFP in the adventitial layer sections from healthy and atherosclerotic aortas (Supplementary Figure 6j,k). Finally, we also showed that vascular macrophages derived from transferred monocytes express CLEC4A2 6 days post DTR-mediated ablation (Supplementary Figure 6f), providing evidence that recruited monocytes can become CLEC4A2⁺ adventitial macrophages in health and disease. The data are included in the Results (Page 9 Line 18).

While these data show that newly recruited monocytes contribute to the reconstituted vascular tissue macrophages, they do not exclude the possibility that local proliferation of resident macrophages also contributed to this process. To address this, we performed new experiments using bone marrow chimeras where the host-derived cells and donor cells were traceable by CD45.1 and CD45.2 expression. CD45.1 hosts reconstituted with bone marrow from *LysM^{Cre} Clec4a2^{flox-DTR}* (CD45.2) mice received a DT or saline injection followed by BrdU injections. The majority of vascular macrophages in the saline group were donor (CD45.2⁺)-derived but a small population of host (CD45.1⁺) cells (radio-resistant) were detectable, which represented approximately 10% of aortic macrophages. In the DT group, proliferation rates were increased from 9.0±1.2% to 43.9±1.9% in the donor origin and from 1.2±0.3% to 8.3±0.8% in the host cells (Supplementary Figure 6l,m). These additional experiments show that the replenished resident macrophage pool includes both origins, but with a predominant contribution of donor cells. The data are included in the Results (Page 10 Line 4).

Overall, our findings are in line with previous studies by Ensan *et al.* (DOI:10.1038/ni.3343) and Lin *et al.* (DOI:10.1172/jci.insight.124574) that already showed the contribution of monocytes to the aortic macrophage pool in health and disease.

Supplementary Figure 6 (related to Figure 2). Replenishment of vascular macrophages by $Ly6C^+$ monocytes post-ablation using $LysM^{Cre+} Clec4a2^{flox/DTR}$ mice. a, Enrichment of $Ly6C^+$ monocytes from the bone marrow (BM) of $Cx3Cr1^{eGFP+} CD45.1^+CD45.2^+$ mice using magnetic bead

negative isolation. **b**, Schematic of diphtheria toxin (DT)-mediated macrophage ablation and adoptive transfer of monocytes in $LysM^{Cre+} Clec4a2^{flox/DTR}$ mice. **c**, Schematic of DT-mediated macrophage ablation and adoptive transfer of monocytes in atherosclerotic mice. BM cells from $LysM^{Cre+} Clec4a2^{flox/DTR}$ mice were transplanted into $Ldlr^{-/-} CD45.1^{+}$ mice. Chimeric mice received an HFD for 12 weeks. Mice were subject to a DT injection followed by adoptive transfer of monocytes. **d, e**, Detection of transferred monocytes in the blood of healthy and atherosclerotic mice by flow cytometry. The mean fluorescent intensity (MFI) of CLEC4A2 expression in monocytes from the host and donor mice. FMO stands for fluorescence minus one. **f, g**, Detection of transferred monocytes in the whole aorta of healthy and atherosclerotic mice by flow cytometry. MFI of CLEC4A2 expression in macrophages from the host and donor mice. **h, i**, Detection of transferred monocytes using immunostaining for CD64 and GFP in the femoral artery of $LysM^{Cre+} Clec4a2^{flox/DTR}$ mice and the ascending aorta of the $Ldlr^{-/-}$ chimeric mice (n=4 mice; Scale bar: 100 μ m). **j, k**, Immuno-staining for CD64 in the femoral artery of $LysM^{Cre+} Clec4a2^{flox/DTR}$ mice and the ascending aorta of the $Ldlr^{-/-}$ chimeric mice, showing depletion and replenishment of adventitial macrophages (n=4 mice; Scale bar: 100 μ m). **l**, Schematic of macrophage depletion and proliferation assays in chimeric $CD45.1^{+}$ mice reconstituted with the BM from $LysM^{Cre+} Clec4a2^{flox/DTR}$ mice. **m**, Percentages of BrdU incorporation in vascular macrophages derived from donor ($CD45.2$) and host ($CD45.1$) origins post saline or DT treatment (n=3 each group).

11) Neutrophils are being depleted in BM and blood for at least 30 hours. Their nb in blood is still substantially lower compared to saline (and also compared to DT-treated $LysM$ -WT/ $Clec4a2^{flox/DTR}$ mice?) at one week of DT treatment. No information is provided beyond this first week. Neutrophils may well be depleted continuously, not only in blood, but also in BM and spleen, with continuous DT treatment.

To better address the kinetics of depletion and repopulation of neutrophils across a variety of organs in $LysM^{Cre+} Clec4a2^{flox/DTR}$ mice, we monitored the effect of DT treatment at additional timepoints (48-hour, 72-hour, 1-week and 2-week) in the bone marrow, blood, spleens and aortas (now included in Figure 2e; Supplementary Figure 5d-i). Neutrophil numbers were fully recovered by 48 hours post-ablation in the bone marrow. Non-statistically significant changes in neutrophil numbers are found in the blood and spleens. The chronic DT injection regimes did not further alter neutrophil counts in these tissues, as previously shown (Supplementary Table 3).

It is important to highlight that neutrophils in the healthy and atherosclerotic aortas were not affected by DT treatments by a single or chronic injection at any time point (Figure 2e; Figure 3b). Although there is short-term neutrophil depletion in the BM, it should be also noted that depletion of neutrophils has been shown to reduce the progression of atherogenesis (DOI:10.1038/s41586-019-1167-6). Therefore, any effect on neutrophil depletion is more likely to underestimate (rather than overestimate) the pro-atherogenic effect of ablation of CLEC4A2⁺ macrophages. This point on that has been added in Discussion (Page 17 Line 23).

Figure 2. *LysM^{Cre+} Clec4a2^{flox/DTR}* mice allow selective ablation of vascular CLEC4A2⁺ macrophages.

a, Representative images of CD64 (red)-stained thoracic aorta sections from *LysM^{Cre+} Clec4a2^{flox/DTR}* and *Clec4a2^{flox/DTR}* mice 24 hours after a saline or diphtheria toxin (DT) injection (n=4 each; Scale bar: 100µm). **b, c, d, e**, Cell numbers of CLEC4A2⁺ macrophages, Ly6C⁺ monocytes, CD11c⁺ macrophages and neutrophils in whole aortas of *LysM^{Cre+} Clec4a2^{flox/DTR}* mice at 1, 2, 3, 5, 7, 14 and 70 days post administration of DT. Day 0 represents the saline-treated control group (n=3-9 each group; pooled from two to three independent experiments). **f**, The mean fluorescent intensity (MFI) of CLEC4A2 expression in vascular macrophages post depletion. FMO stands for fluorescence minus one. **g-l**, *LysM^{Cre+} Clec4a2^{flox/DTR}* and *Clec4a2^{flox/DTR}* mice received three intraperitoneal injections of saline or DT in a week. Aortic cells were analysed by flow and mass cytometry. **g-j**, Representative flow cytometry plots depict macrophages, Ly6C⁺ monocytes, cDC2 and neutrophils from aortas of saline or DT-treated *LysM^{Cre+} Clec4a2^{flox/DTR}* mice and DT-treated *Clec4a2^{flox/DTR}* mice. Cell numbers of

each population in the three groups. The data are representative of three independent experiments. **k**, Overlaid tSNE plots of vascular leucocytes from saline or DT-treated *LysM^{Cre+} Clec4a2^{fllox/DTR}* mice by mass cytometry. Leucocyte populations were identified based on expression of 32 markers. **l**, Cell numbers in 9 aortic leucocyte populations across the 2 treatment groups by mass cytometry. Data are representative of two independent experiments. All data are presented as mean±SEM. Ordinary one-way ANOVA with Holm-Sidak's multiple comparisons test or Two-tailed Student's t-test. *P<0.05, **P<0.01, ***P<0.001, ****P<0.0001.

Supplementary Figure 5. (related to Figure 2).

a, The gene construct for tracing and ablating cells expressing both *LysM* and *Clec4a2*. **b**, Endogenous expression of eGFP in neutrophils and Ly6C⁺ monocytes in the bone marrow (BM) and blood of *Clec4a2*^{flox/DTR} and *LysM*^{Cre+} *Clec4a2*^{flox/DTR} mice. **c**, Gene expression of *Clec4a2*, *egfp* and *Hbegf* (*DTR*) in sorted macrophages (CD45⁺CD11b⁺CD64⁺F4/80⁺), Ly6C⁺ monocytes (CD45⁺CD11b⁺Ly6C⁺CD64⁺) and dendritic cells (DCs) (CD45⁺CD11c⁺MHCII⁺CD64⁺) from the whole aorta and lungs of *LysM*^{Cre+} *Clec4a2*^{flox/DTR} mice by RT-qPCR. **d-i**, Percentages of neutrophils and Ly6C⁺ monocytes from the BM, blood and spleen of *LysM*^{Cre+} *Clec4a2*^{flox/DTR} mice at 6, 24, 30, 48, 72 hours and 1- and 2-weeks post a single DT injection. Data are representative of two or three independent experiments. **j,k**, Representative flow cytometry plots showing macrophage depletion and cell numbers in the heart and lungs of *LysM*^{Cre+} *Clec4a2*^{flox/DTR} and *Clec4a2*^{flox/DTR} mice that received three intraperitoneal injections of saline or diphtheria toxin (DT) in a week. All data are presented as mean±SEM. Ordinary one-way ANOVA with Holm-Sidak's multiple comparisons test. *P<0.05, **P<0.01, ***P<0.001, ****P<0.0001.

12) Does the killing of macrophages or neutrophils induce “non-specific” inflammation (unrelated to or super-imposed on to the atherosclerosis process), unrelated to CLEC4A2 function or to the functions of those cells? (Necrotic) cell death-induced inflammation? Is there any sign of systemic inflammation due to the killing of these cells?

Please refer to Point 1 which addresses this comment.

13) Figure 2: Loss of resident macrophages should lead to vessel dilatation, according to previous work. Have you seen this?

We do not observe significant expansion of the artery after CLEC4A2⁺ macrophage depletion, possibly due to the relatively short timecourse and the selective targeting of a smaller macrophage population, compared to previous findings of arterial enlargement in WT mice with a CSF1R inhibitor treatment for 14 weeks (DOI:10.1016/j.immuni.2018.06.008).

However, we observed arterial expansion in *ApoE*^{-/-} *Clec4a2*^{-/-} mice which had significantly reduced resident vascular macrophages (Figure 4d), further supporting our conclusion that the effects of the pan-cellular KO are linked to the effects of the resident vascular macrophages.

14) The collar model is not analysed in detail. It's unclear what contributed to the increase of neointima formation.

We agree that additional analysis can be done to better explore these aspects. However, having included significant amount of new data in resubmission, we prefer to focus the study on the exploration of the role of CLEC4A2 in atherosclerosis. Therefore, we removed the data on the collar model to make space for the revision data.

15) BM transplant atherosclerosis experiment: irradiation will have depleted resident macrophages. Do those resident macrophages reconstitute (in the adventitia) from circulating monocytes before the start of HFD?

Yes, adventitial macrophages are fully reconstituted 4 weeks after irradiation and BM transfer. All chimera mice used in this manuscript were recovered for at least 5 weeks post irradiation and transfer before HFD feeding.

16) Again, it's important to see data on what chronic killing of a huge number of cells throughout the body is doing to the general inflammatory response.

Please refer to Point 1 which addresses this issue.

17) Plasma lipid levels are not provided.

Thanks for pointing this out. This submission contains the cholesterol data which show that systemic cholesterol levels were not significantly altered by ablation of CLEC4A2⁺ macrophages (now added Supplementary Table 4). These data are discussed in the Results (Page 10, Line 25) and in the Discussion (Page 18, Line 4).

18) Atherosclerosis was assessed only in en-face aorta.

We performed additional analysis exploring lesion size in aortic roots and ascending aortas using Oil Red O staining and the data consistently showed that lesion formation was larger in mice with macrophage depletion (Figure 3e-h; see below). This supports our data from the matching *en face* aortic preparations that depletion of CLEC4A2⁺ macrophages leads to an increase in atherosclerosis.

Figure 3. CLEC4A2⁺ macrophages protect from atherosclerosis.

a, Schematic of macrophage ablation in atherosclerotic mice. Bone marrow (BM) cells from $LysM^{Cre+}$ $Clec4a2^{flox/DTR}$ and $Clec4a2^{flox/DTR}$ mice were transplanted into $Ldlr^{-/-}$ (CD45.1) mice. The chimeric mice received an HFD for 12 weeks and intraperitoneal injections of saline or DT fortnightly for 12 weeks. **b**, Cell numbers of resident macrophages, $CD11c^+$ macrophages, $Ly6C^+$ monocytes, $Ly6C^-$ monocytes, $cDC2$ and neutrophils from aortas for each treatment group. **c, d**, Representative images and plaque quantification of Sudan IV (red)-stained en face whole aortas of $Ldlr^{-/-}$ chimeric mice for each treatment group (Scale bar: 5mm). **e, f**, Representative images and plaque quantification of Oil Red O (ORO)-stained sections of ascending aorta of $Ldlr^{-/-}$ chimeric mice for each treatment group (Scale bar: 200 μ m). **g, h**, Representative images and plaque quantification of ORO-stained aortic root sections of $Ldlr^{-/-}$ chimeric mice for each treatment group (Scale bar: 200 μ m). All data are presented as mean \pm SEM. Ordinary one-way ANOVA with Holm-Sidak's multiple comparisons test. * $P < 0.05$, ** $P < 0.01$, *** $P < 0.001$.

19) Figure 4: Here the experiments involve whole KO of *Clec4a2*. Therefore, they are not specific for macrophages. A better experiment would be *LysM-cre/Clec4a2* floxed mice crossed to *apoe*^{-/-}. The increase of lesion size in *Clec4a2* KO cannot be attributed to changes in macrophage subsets or function.

The difference in aortic immune cells subsets between the 2 groups of mice could just reflect the difference in lesion size at the end of the experiment. Indeed, progression of atherosclerosis is known to be associated with reduced M2-like macrophages and increased M1-like macrophages.

*Also, the use of *Lyve1-cre/Clec4a2* floxed mice could be more appropriate to address the role of *Clec4a2* more specifically in resident-like macrophages.*

Please refer to Point 2 which addresses this issue.

20) Figure 5: It would be informative to get these BMDM stimulated using oxLDL or other atherosclerosis-relevant stimuli and assess how *Clec4a2* deficiency would impact on their response/phenotype.

To directly address this comment, we performed new experiments using BMDM from WT and KO mice. We tested IL-6 secretion in CSF1-cultured BMDMs with oxLDL stimulation and found that IL-6 release after exposure to oxLDL did not differ between genotypes (see below).

CSF1-cultured BMDMs from WT and $Clec4a2^{-/-}$ mice were treated with oxLDL for 48 hours and IL-6 levels in the supernatant were measured using ELISA. Data are presented as mean \pm SEM. Ordinary one-way ANOVA with Holm-Sidak's multiple comparisons test. * $P < 0.05$.

21) Figure 6: these experiments indicate that the deletion of *Clec4a2* in all BM-derived cells could impact not only macrophages but also DC numbers and functions. Thus, the experiment in Figure 4 doesn't allow for any valid conclusion to be drawn regarding the contribution of macrophage-*Clec4a2* to the observed atherosclerotic phenotype. Experiments in Figure 4 and 6 are better done using *LysM-cre/Clec4a2^{flox}* mice.

Please refer to Point 2 which addresses this issue.

The competitive chimera experiments (Figure 6) and new experiments (Figure 5f,h; Supplementary Figure 8), together with the data from transcriptional studies (that confirm the reciprocal changes in CSF1 and CSF2 pathways both *in vitro* and *in vivo*), support the concept that there is a single overarching mechanism for the remodelling of the aortic myeloid compartment in the absence of CLEC4A2.

Additionally, it is worth noting that the DT-mediated cell ablation in *LysM^{Cre} Clec4a2^{flox-DTR}* mice exclude DCs from the ablation.

22) Figure 7: The finding that deletion of *Clec4a2* affects only one subset out of 3 of 4 subsets expressing *Clec4a2* is not highly supportive of a critical role of *Clec4a2* in the establishment of a resident-like macrophage phenotype.

We apologise for not sufficiently clarifying the changes in macrophage subsets that specifically relate to CLEC4A2 deletion. In Figure 7, scRNA-seq data describe changes between CLEC4A2 competent and deficient *ApoE^{-/-}* mice. In this setting, only 2 Lyve1⁺ macrophage subsets are identified, compared to the 3 subsets identified in Figure 1. In absence of CLEC4A2, there is a significant decrease in resident vascular macrophages (Cluster 1) that have a non-activated resident transcriptional signature (Figure 7b). The clustering algorithm does not identify CD209⁺Lyve1⁺ macrophages as a separate cluster in the dataset in Figure 7 (compared to Figure 1), possibly due to the considerable decrease of these subsets in *ApoE^{-/-} Clec4a2^{-/-}* mice, as evident in the mass cytometry data (Figure 4). This clarification has now been added to the Results (Page 15, Line 3).

It is worth highlighting that the selectivity of expression of CLEC4A2 within resident macrophages is a specific advance in the field that our manuscript offers, and not a drawback. We show that there is significant heterogeneity in Lyve1⁺ macrophage transcriptional states and that a significant proportion of Lyve1⁺ macrophages acquire an activated phenotype during atherogenesis and they do not express CLEC4A2. CLEC4A2 expression distinguishes activated Lyve1⁺ macrophages (Cluster C0 in both Figure 1 and Figure 7) from non-activated Lyve1⁺ macrophages (Clusters C1 and C5 in Figure 1 and Cluster C1 in Figure 7), as acknowledged by the Reviewer in point 6. CLEC4A2 deletion only reduces the representation of the CLEC4A2⁺ macrophage subset (C1 in Figure 7), as it would be consistent with its expression.

23) I'm wondering whether you would see the same difference in immune cell distribution just by comparing plaques of different size (at different stage of development).

One way to show that *Clec4a2* deficiency directly affects the subsets of macrophages within the lesions is to repeat the experiment at an earlier time point at which no difference in lesion size can be detected between the 2 groups.

Please refer to Point 2 which addresses this issue in full.

24) Would the finding that *Clec4a2* deficiency promotes both foam cell formation and a pro-inflammatory phenotype be counter-intuitive knowing that most foam cell macrophages in atherosclerosis display an anti-inflammatory phenotype?

Our scRNA-seq data showed that the increase in lesion size in CLEC4A2 deficiency is associated with an increase of two *Itgax* (CD11c) expressing macrophage subsets: inflammatory monocyte-macrophages (Cluster 2) and Trem2-expressing macrophages (Cluster 6) (Figure 7b). Trem2⁺ macrophages have a transcriptomic profile that is consistent with foam cells lacking inflammatory genes, as the Reviewer points out.

Reviewer #2 (Remarks to the Author):

This manuscript by Monaco and colleagues identifies the C-type lectin CLEC4A2 as a defining marker of vascular resident macrophages and demonstrates its essential role in deciding the fate of monocytes as they differentiate in situ. The role of CLEC4a2 in both vascular homeostasis and disease (arterial injury and atherosclerosis) is investigated using macrophage-specific deletion (LysMCreClec4a2flox-DTR) and whole body knock-out mouse models through a combination of mass cytometry, scRNA-seq and conventional histological analyses. The work is truly a tour de force and the manuscript is clearly written. The authors should be commended for their clear distillation of complicated datasets, and the integration of multiple independent techniques to demonstrate the role of CLEC4a2. I have only minor suggestions to improve the clarity of the results.

We are extremely grateful to the Reviewer for recognising the careful nature of our work and our use of multiple independent techniques to support our conclusions.

1. The authors begin by defining the macrophage populations in the intima-media and adventitial layers of aortas of ApoE^{-/-} mice, and demonstrate that the majority of CD206⁺CD169⁺ macrophages reside in the adventitia, while CD11c⁺CD44⁺ macrophages predominantly reside in the intima. In subsequent studies, this division is no longer addressed and the reader is left to guess that all further studies use the whole aorta. This should be clarified.

We apologise for not clarifying these important details in the manuscript. In Supplementary Figure 1, we used mass cytometry and immunofluorescence microscopy to identify which subsets of macrophages are placed in the intima and adventitia using a partial digestion to separate the intima/media from the adventitia. Separating the intima and the adventitia requires an additional enzymatic digestion process resulting in loss of cells and a high number of experimental animals are needed to compensate for cell loss. We were unable to use this approach throughout the manuscript. However, there were clear differences in the immune landscape between the two compartments. Once we identified the location of the subsets with the separation technique, we applied these markers to characterise aortic macrophage populations in the whole aorta. To avoid confusion, we clarified in the Figure Legends where the whole aorta was used.

2. In the results section describing LysMCre-Clec4a2flox-DTR mice, the expression of eGFP in various immune cell subtypes is described without any previous mention of how eGFP is incorporated into the LysMCre-Clec4a2flox-DTR mice or what it is reporting. This should be clarified.

We apologise that the information of the mouse strain construct was not clearly stated. We have now included further clarification in the Results (Page 8, Line 7) and highlighted in the Method sections (Page 27, Line 15).

3. On page 10, the sentence that begins “The most notable change caused by CLEC4A2...” needs to be rewritten.

Thank you for the careful read and for pointing out the lack of clarity in this sentence. This has now been amended (Page 12, Line 5; pasted below this response for the Reviewer’s convenience).

“When mice developed advanced disease following feeding of an HFD, we consistently found a significant decrease in resident macrophages (**Figure 4j,k**). In contrast to the earlier stages of atherogenesis, an increase of CD11c⁺ macrophages, Ly6C⁺ monocytes and neutrophils was detected in the aorta of HFD-fed *ApoE^{-/-} Clec4a2^{-/-}* mice (**Figure 4j,k**).”

4. In the discussion, the authors provide a potential mechanism through which CLEC4a2 may regulate CSF2-mediated fate decisions, via its interaction with SHP1. Studies of SHP1^{-/-} macrophages showing that they phenocopy Clec4a2^{-/-} macrophages would have strengthened this argument. Given the large amount of data already in the manuscript, these experiments are not required, but would have helped close the “mechanistic loop”.

We thank the Reviewer for this valuable suggestion. The *motheaten* mouse strain (SHP1^{-/-}) shares many similarities with CLEC4A2 deficient mice, including inhibition of TLR signalling and CSF2 pathways (DOI:10.1006/cimm.1998.1272 and 10.1016/j.immuni.2013.02.018), and susceptibility to autoimmune disease (DOI:10.1038/ng0693-124). This is included in the Discussion (Page 21, Line 8). However, as SHP1 is widely expressed and acts downstream to a number of immune-regulatory pathways, the phenotype of the *motheaten* mouse is more severe and systemic than *Clec4a2^{-/-}* mice. We attempted to obtain a mouse strain with a conditional deletion of SHP1 in myeloid cells to answer the Reviewer’s comment but we were unsuccessful in our several attempts of acquiring the mouse in time for this revision. We agree that this is a new direction that will need to be explored in future experiments.

Reviewer #3 (Remarks to the Author):

Previously the authors discovered two major macrophage subsets in the atherosclerotic aorta of mice: CD206⁺CD169⁺ and CD11c⁺CD44⁺. In this follow up study, the authors find that CD206⁺CD169⁺ abundance is lower in atherosclerotic aortas compared to WT and that Clec4a2 critically modulates this process. Using single cell RNA seq, their initial clue into this process is that Clec4a2 expression in macrophages from atherosclerotic aortas is negatively correlated with an inflammatory gene signature. Using a Cre-floxed DTR system to selectively ablate Clec4a2 macrophages and partial BM chimera to KO Clec4a2, the authors found that atherosclerotic lesions are significantly larger and more necrotic in the absence of Clec4a2. Further, in the absence of Clec4a2, macrophages were: i) more likely to differentiate into Ly6c⁺ macrophages, ii) sensitized to TLR stimulation; iii) lacking lipid metabolism genes; iv) poorly equipped for cholesterol efflux. The authors deduce that Clec4a2 promotes monocyte differentiation into resident macrophages in healthy and atherosclerotic aortas and thus protects the aorta from atherosclerotic lesions.

Noteworthy results include:

- 1. Clec4a2 expression is inversely associated with an inflammatory signature in scRNA-seq.**

2. **Atherosclerotic lesions are bigger and more necrotic in the absence of Clec4a2+ macrophages,**
3. **Clec4a2 expression biases differentiation: Clec4a2 +/+ (WT) more likely to differentiate into res macs; Clec4a2 -/- more likely to differentiate into CD11c, Ly6c, cDC2.**
4. **Macrophages missing Clec4a2 downregulate lipid metabolism genes and thus may poorly process lipids or export cholesterol, possibly contributing to larger lesions.**
5. **Macrophages missing Clec4a2 are sensitized and express more IL6 in response to TLR ligands**

1) The work in this paper is of very high quality and supports the majority of the conclusions. There is enough supporting information in the methods for reproduction (except I could not find the 170-gene inflammatory signature).

We thank the Reviewer for the positive comments on our manuscript. We have respectfully numbered each point of the reviewer for clarity.

We apologise for this omission in the initial submission. We have now added the list in Supplementary Table 2.

2) As for significance, their finding that Clec4a2 expression is negatively correlated with an inflammatory signature is critical, and led them to delete Clec4a2 in macrophages and discover an important but previously unknown role for Clec4a2 in tissue macrophage homeostasis. Their results are significant for the field of atherosclerosis and macrophage biology and could also impact numerous areas of biology if Clec4a2 has a similar role in other damaged organs and if the results extend to human disease.

We are extremely grateful to the Reviewer for the very positive comments on our work.

Criticisms:

3) After culturing CSF1- and CSF2-BMDM from WT and KO mice ex vivo, it wasn't clear what they differentiate into. Do the differentiated states match those identified in vivo from KO vs WT BM transfers? Are the "core" gene sets identified in vivo similar in the cultured cells?

We thank the Reviewer for raising the important point of the convergence between the *in vivo* and *in vitro* effects of CLEC4A2 deficiency. We added the representative images of gating strategy for macrophages in CSF1 and CSF2 culture conditions to exemplify the acquired phenotype *in vitro* (added in Figure 5e,g): CSF1-BMDMs: CD11b⁺ F4/80⁺ CD206⁺; CSF2-BMDMs :CD11b⁺ CD11c⁺ F4/80⁺ MHCII⁺. The marker profile *in vitro* matches with the profile *in vivo* overall.

To investigate how CLEC4A2 affects macrophages *in vitro* and *in vivo*, we performed bulk RNA-seq in the *in vitro* experiments (Figure 5) and scRNA-seq within the aortic wall (Figure 7). We agree that each tissue shapes a specialised transcriptional profile in macrophages. Therefore, the *in vitro* culture setting does not fully replicate the vascular microenvironment that influences maturation of macrophages. Also comparing bulk and scRNA-seq findings has its challenges, due to the more limited number of gene per cell identified by scRNA-seq. Nevertheless, we did find a significant number of common genes regulated by CLEC4A2 between CSF1-cultured macrophages (Figure 4a,b) and resident vascular macrophages (Cluster 1 from Figure 7a-c). We now generated a table highlighting the overlapping genes (added to this revision in Supplementary Table 11). The existence of this overlap in CLEC4A2-dependent transcriptional profiles *in vitro* and *in vivo* supports the validity of our

data and overall conclusions. We have now better highlighted the similarities of the *in vitro* and *in vivo* transcriptional findings in the Results (Page 15 Line 18) and Discussion (and Page 19 Line 3).

Supplementary Table 11. A list of differentially expressed genes (P<0.05) by CLEC4A2 deficiency that are found in both CSF1-BMDMs (*Clec4a2*^{-/-} vs. WT mice) and vascular resident macrophages (Cluster 1; *ApoE*^{-/-} *Clec4a2*^{-/-} vs. *ApoE*^{-/-} mice).

Downregulated	Upregulated
Ccr3	Apobec1
Prss23	Mosmo
Srp54a	Il7r
Bzw1	Snx10
Srp54c	Snx15
Clec4b1	Jade2
Tpbgl	Depp1
Tmem8	Rgs1
1700047117Rik2	Pi4k2b
Clec4a4	Cd1d1
Cat	4930503L19Rik
Trim30a	Mbd4
Srp54b	Plxnd1
Clec4a2	Rbp1
Znf41-ps	Vamp2
Sema6d	F10
Zfp992	Naip6
Tmf1	
Ide	
Lpp	
Ids	
Cyp51	
G6pd2	
Ddx59	
Gosr2	
Atp6v1e1	
Fam177a	
Trib2	
Gpr183	
G530011006Rik	
Gal	
Alox5	

4) As the figure and text stands, one interpretation of these data is that CSF1 induces WT cells to proliferate, and that CSF2 induces *Clec4a2*^{-/-} cells to proliferate. But the text does not make it clear whether these cells differentiate into a relevant state or express (or lack) gene programs identified *in vivo*. Thus, while this is an interesting observation *ex vivo*, the results do not link up yet with the results *in vivo*. Still, the finding that CSF2-induced TFs are upregulated *in vivo* (as mentioned in discussion) does provide some more support for the theory. Future experiments with CSF2 deficient mice, or other manipulations of CSF1 and 2, may help clarify these questions.

We are extremely grateful to the Reviewer for the excellent suggestion to look into proliferation as a mechanism for CLEC4A2-dependent differentiation bias. We performed new experiments to investigate whether proliferation in macrophages is affected in the absence of CLEC4A2 *in vivo* and *in vitro* using EdU/BrdU incorporation assays. When bone marrow-derived macrophages from WT and KO mice were cultured in the presence of CSF1 or CSF2, WT cells proliferated more efficiently than the KO counterpart in the CSF1-culture whilst KO macrophages were highly proliferative in the CSF2-dependent culture (added in Figure 5f,h; see below). These results are consistent with the Reviewer's interpretation.

We also agree with the Reviewer that it is important to link up the results *in vitro* and *in vivo* for the reasons discussed already at Point 3 above. In order to assess whether CLEC4A2 could induce a proliferation bias in the artery wall as well, we measured proliferation *in vivo* within the arterial wall both in health and atherosclerotic disease. In the healthy aorta, proliferation rate was not altered in resident vascular macrophages from *Clec4a2*^{-/-} mice relative to the WT (added in Supplementary Figure 8 a-c; see below). This is consistent with our earlier findings that CLEC4A2 deficient mice have a normal numerical complement of vascular resident macrophages at the steady state (Supplementary Table 5). However, when atherosclerotic lesions were established and two macrophage compartments were formed, CLEC4A2 deficiency resulted in a reduction in proliferation rates in resident vascular macrophages whilst enhancing proliferation in CD11c⁺ macrophages (Supplementary Figure 8d-h).

These new results, together with the matching evidence from the *in vitro* maturation assays (Figure 5e,g) and the *in vivo* competitive chimera (Figure 6), and the matching transcriptional evidence *in vivo* and *in vitro* (see discussion of point 3 above), support the conclusion that CLEC4A2 biases macrophage differentiation towards the preservation of the resident vascular macrophage pool and its essential homeostatic properties.

We apologise that we were not able to obtain CSF1/2 KO mice within the timeframe to replicate the phenotype of CLEC4A2 deficient macrophages. We agree that future experiments on the role of CSF1 and CSF2 in the setting of vascular macrophage biology would be of significant interest.

Figure 5. CLEC4A2 maintains macrophage identity and homeostasis by promoting CSF1- over CSF2-dependent differentiation *in vitro*.

a, Principal component analysis (PCA) of the RNA-seq data from CSF1-BMDMs of WT and *Clec4a2*^{-/-} mice. **b**, Heatmap of differentially expressed genes (P<0.05) between WT and *Clec4a2*^{-/-} CSF1-BMDMs by RNA-seq. (n=3 mice, the heatmap represents one of two independent experiments). **c**, Expression of CD206 in WT and *Clec4a2*^{-/-} CSF1-BMDMs by flow cytometry (n=3 mice; two independent experiments). **d**, Schematic showing co-culture of BM cells from 50 % WT (CD45.1/CD45.2 heterozygous) and 50 % *Clec4a2*^{-/-} (CD45.2) mice with CSF1. **e**, Representative flow cytometry plots and proportion of each origin in CD11b⁺F4/80⁺CD206⁺ CSF1-BMDMs. (n=3 mice; two independent experiments). **f**, Representative flow cytometry plots and percentages of BrdU incorporation in co-cultured BMDMs (CD11b⁺F4/80⁺CD206⁺) with CSF1 (n=3 mice). **g**, BM cells of 50

% WT (CD45.1) and 50 % *Clec4a2*^{-/-} (CD45.2) mice were co-cultured with CSF2. Adherent cells were analysed by flow cytometry. Representative flow cytometry plots and proportion of each origin in CD11b⁺CD11c⁺F4/80⁺MHCII⁻ CSF2-BMDMs. (n=4 mice; two independent experiments). **h**, Representative flow cytometry plots and percentages of BrdU incorporation in co-cultured BMDMs (CD11b⁺CD11c⁺F4/80⁺MHCII⁻) with CSF2 (n=3 mice). All data are presented as mean±SEM. Two tailed Student's t test; *P<0.05, **P<0.01.

Supplementary Figure 8 (related to Figure 6).

a, Schematic of proliferation assays in WT and *Clec4a2*^{-/-} mice. BrdU was injected intraperitoneally every other day for 9 days. **b**, **c**, Representative images and percentages of BrdU incorporation in CD206⁺Lyve1⁺ resident vascular macrophages. **d**, Schematic of proliferation assays in *ApoE*^{-/-} and *ApoE*^{-/-} *Clec4a2*^{-/-} mice fed an HFD for 12 weeks. BrdU was injected intraperitoneally every other day for 9 days before sacrifice. **e**, **f**, **g**, **h**, Representative images and percentages of BrdU incorporation in CD206⁺Lyve1⁺ resident vascular macrophages and CD206⁺Lyve1⁺CD11c⁺ macrophages. All data are presented as mean±SEM. Two tailed Student's t test. *P<0.05.

REVIEWER COMMENTS

Reviewer #1 (Remarks to the Author):

No further comments

Reviewer #2 (Remarks to the Author):

The reviewers have satisfied my previous concerns and have made a good faith effort to address all of the reviewers' critiques. However, some of the new data added in Figure 3 have raised additional concerns about the size of experimental groups used for atherosclerosis analyses. Studies in Figure 3e-h are not sufficiently powered to allow analysis of lesion area in the aortic root given known variation in the measurement. Most power analyses indicate a minimum group size of 8 mice for such measures, and the data shown comes from groups of 5-6 mice.

Reviewer #3 (Remarks to the Author):

My comments were adequately experimentally addressed by authors and clarified key points. First, the authors provide acceptable additional evidence that their in vivo system recapitulates to some extent the in vivo cellular phenotypes they observed by identifying shared genes regulated by Clec4a2 in cells collected from the aorta and in culture as assessed by scRNA-seq or bulk RNA seq (they also monitored identity markers using flow from cells collected in vivo and in vitro). Second, they provided new in vitro evidence that Clec4a2 biases the monocyte to macrophage transition in response to CSF1 by monitoring proliferation in WT and Clec4a2 ^{-/-} cells. Together, their responses help link their in vitro and in vivo results to provide (albeit limited) mechanistic insight into how Clec4a2 protects against atherosclerosis.

POINT-BY-POINT RESPONSE TO THE REVIEWERS**Reviewer #1 (Remarks to the Author):**

No further comments.

We thank the Reviewer again for his/her comments from the initial submission, which improved the quality of our work.

Reviewer #2 (Remarks to the Author):

The reviewers have satisfied my previous concerns and have made a good faith effort to address all of the reviewers' critiques. However, some of the new data added in Figure 3 have raised additional concerns about the size of experimental groups used for atherosclerosis analyses. Studies in Figure 3e-h are not sufficiently powered to allow analysis of lesion area in the aortic root given known variation in the measurement. Most power analyses indicate a minimum group size of 8 mice for such measures, and the data shown comes from groups of 5-6 mice.

We would like to thank the Reviewer for his/her support throughout the revisions that helped us improve considerably the manuscript. We apologise for the insufficient experimental size for atherosclerosis analyses, which was due to time constraints. As suggested by the Reviewer, we performed further oil red O (ORO) staining on aortic roots from additional mice, increasing the sample size to 9-11 per group. The new data consistently show that lesional areas in aortic roots and ascending aortas were larger in mice deficient in CLEC4A2⁺ macrophages (Figure 3f,h or see below). These findings support our conclusion that CLEC4A2⁺ macrophages are required for preventing severe atherosclerosis.

Figure 3. CLEC4A2⁺ macrophages protect from atherosclerosis.

a, Schematic of macrophage ablation in atherosclerotic mice. Bone marrow (BM) cells from *LysM^{Cre+} Clec4a2^{fllox/DTR}* and *Clec4a2^{fllox/DTR}* mice were transplanted into *Ldlr^{-/-} (CD45.1)* mice. The chimeric mice received an HFD for 12 weeks and intraperitoneal injections of saline or DT fortnightly for 12 weeks. **b**, Cell numbers of resident macrophages, CD11c⁺ macrophages, Ly6C⁺ monocytes, Ly6C⁻ monocytes, cDC2 and neutrophils from aortas for each treatment group. **c**, **d**, Representative images and plaque quantification of Sudan IV (red)-stained en face whole aortas of *Ldlr^{-/-}* chimeric mice for each treatment group (Scale bar: 5mm). **e**, **f**, Representative images and plaque quantification of Oil Red O (ORO)-stained sections of ascending aorta of *Ldlr^{-/-}* chimeric mice for each treatment group (Scale bar: 200μm). **g**, **h**, Representative images and plaque quantification of ORO-stained aortic root sections of *Ldlr^{-/-}* chimeric mice for each treatment group (Scale bar: 200μm). All data are presented as mean±SEM. Ordinary one-way ANOVA with Holm-Sidak's multiple comparisons test. *P<0.05, **P<0.01, ***P<0.001. Source data are provided as a Source Data file.

Reviewer #3 (Remarks to the Author):

My comments were adequately experimentally addressed by authors and clarified key points. First, the authors provide acceptable additional evidence that their in vivo system recapitulates to some extent the in vivo cellular phenotypes they observed by identifying shared genes regulated by Clec4a2 in cells collected from the aorta and in culture as assessed by scRNA-seq or bulk RNA seq (they also monitored identity markers using flow from cells collected in vivo and in vitro). Second, they provided new in vitro evidence that Clec4a2 biases the monocyte to macrophage transition in response to CSF1 by monitoring proliferation in WT and Clec4a2 -/- cells. Together, their responses help link their in vitro and in vivo results to provide (albeit limited) mechanistic insight into how Clec4a2 protects against atherosclerosis.

We would like to thank the Reviewer for the positive comments on our additional work and for the excellent summary. Indeed, the consistency of our data *in vitro* (Figure 5f,g) and *in vivo* (Supplementary Figure 8) showing an effect on transcriptional changes and proliferation supports a role for the CLEC4A2 pathway in monocyte to macrophage transition. Future studies will help mine further the mechanisms underlying CSF1/CSF2 regulated events in vascular macrophages.

We altered the discussion accordingly to clarify this concept (Page 18 Line 20)

REVIEWERS' COMMENTS

Reviewer #2 (Remarks to the Author):

The authors have satisfied all remaining concerns

POINT-BY-POINT RESPONSE TO THE REVIEWERS

Reviewer #2 (Remarks to the Author):

The authors have satisfied all remaining concerns

We thank the Reviewer again for his/her support throughout the revisions that helped us improve considerably the manuscript.